# STRUCTURE-PRESERVING CONTRASTIVE LEARNING FOR SPATIAL TIME SERIES

## ABSTRACT

Informative representations enhance model performance and generalisability in downstream tasks. However, learning self-supervised representations for spatially characterised time series, like traffic interactions, poses challenges as it requires maintaining fine-grained similarity relations in the latent space. In this study, we incorporate two structure-preserving regularisers for the contrastive learning of spatial time series: one regulariser preserves the topology of similarities between instances, and the other preserves the graph geometry of similarities across spatial and temporal dimensions. To balance contrastive learning and structure preservation, we propose a dynamic mechanism that adaptively weighs the trade-off and stabilises training. We conduct experiments on multivariate time series classification, as well as macroscopic and microscopic traffic prediction. For all three tasks, our approach preserves the structures of similarity relations more effectively and improves state-of-the-art task performances. This approach can be applied to an arbitrary encoder and is particularly beneficial for time series with spatial or geographical features. Our code is attached as supplementary material, which will be made openly available with all resulting data after review.

## 1 INTRODUCTION

Self-supervised representation learning (SSRL) theoretically can learn latent embeddings that facilitate downstream tasks (Saunshi et al., 2019; HaoChen et al., 2021; Ge et al., 2024). Also, it is practically shown to improve model generalisability (Tendle & Hasan, 2021; Zhou et al., 2022). The latter is particularly valuable for real-world applications, where both measurements and labels are often uncertain and unreliable. In fact, SSRL has been widely applied across fields such as computer vision, natural language processing, and recommendation systems (there are many literature reviews, to name a few, Schiappa et al., 2023; Liu et al., 2023; Yu et al., 2024).

Contrastive learning has become the mainstay technique in SSRL of time series. Lafabregue et al. (2022) conducted an extensive experimental comparison over 300 combinations of network architectures and loss functions to evaluate the performance of time series representation learning for clustering. One of their key findings is that the reconstruction loss used by traditional autoencoders does not sufficiently fit temporal patterns. Instead, contrastive learning has emerged as a more effective approach, which embeds similar samples closer together while dissimilar samples farther apart in the latent space (Wu et al., 2023; Yang et al., 2024).

Unique challenges arise when learning contrastive representations for spatial time series. First, data with both temporal and spatial characteristics demand more fine-grained similarity comparisons, which underpins contrastive learning. Financial time series may be considered similar even if some variables show significant divergence, while movement traces with very different spatial features can be anything but similar. Second, effective representation of spatial time series needs to capture spatio-temporal patterns at the certain scale required by a practical task. For example, traffic interactions involve two different spatial scales: at the macroscopic scale, traffic flow measures collective road usage evolving over the road network; at the microscopic scale, trajectories describe the motion dynamics of individual road users (e.g., car drivers, cyclists, pedestrians) in local road space.

To address the challenges, we incorporate two regularisers *at different scales* to *preserve the original similarity structure* in the latent space for time series contrastive learning. One is a topology-preserving regulariser for the global scale, and the other is a graph-geometry-preserving regulariser

for the local scale. This incorporation can be simplified as a weighted loss $\mathcal{L} = \eta_{\text{CLT}} \cdot \ell_{\text{CLT}} + \eta_{\text{SP}} \cdot \ell_{\text{SP}} + r_{\boldsymbol{\eta}}$, where we propose a mechanism to dynamically balance the weights $\eta_{\text{CLT}}$ and $\eta_{\text{SP}}$ during training. Within this mechanism, the adaptive trade-off between contrastive learning and structure preservation is based on the uncertainties of their corresponding terms $\ell_{\text{CLT}}$ and $\ell_{\text{SP}}$; meanwhile, the term $r_{\boldsymbol{\eta}}$ adds regularisation against overfitting of the dynamic weights.

The proposed approach is applicable to spatial time series in general, and we consider traffic interaction as a specific case. To validate the approach, we conduct experiments on tasks of 1) multivariate time series classification, where we benchmark against the current state-of-the-art (SOTA) models, i.e., TS2Vec Yue et al. (2022) and Lee et al. (2024); and 2) traffic prediction, where we use Li et al. (2024a) for macroscopic benchmark and Li et al. (2024b) for microscopic. In addition, the efficiency of this approach is evaluated with various model architectures. Below we summarise the key contributions of this study:

- We introduce an approach that incorporates structure-preserving regularisation in contrastive learning of multivariate time series, to maintain finer-grained similarity relations in the latent space of sample representations. We propose a dynamic weighing mechanism to adaptively balance contrastive learning and structure preservation during training.

- Preserving similarity structure can enhance SOTA performance on various downstream tasks. The *relative improvement* on spatial datasets in the UEA archive is 2.96% in average classification accuracy; on macroscopic traffic prediction task is 0.72% in flow speed RMSE and 0.27% in explained variance; on microscopic trajectory prediction task is 3.72% and 8.10% in missing rates under radii of 0.5m and 1m, respectively.

- This approach can be applied to an arbitrary encoder for self-supervised representation learning. Preserving the structure of similarity relations is particularly beneficial for time series data with spatial or geographical characteristics, such as in robotics, meteorology, remote sensing, urban planning, etc.

## 2 RELATED WORK

### 2.1 TIME SERIES CONTRASTIVE LEARNING

Contrastive learning for time series data is a relatively young niche and is rapidly developing. The development has been dominantly focused on defining positive and negative samples. Early approaches construct positive and negative samples with subseries within time series (e.g., Franceschi et al., 2019) and temporal neighbourhoods (e.g., Tonekaboni et al., 2021); and later methods create augmentations by transforming original series (e.g., Eldele et al., 2021; 2023). More recently, Yue et al. (2022) generates random masks to enable both instance-wise and time-wise contextual representations at flexible hierarchical levels, which exceeds previous state-of-the-art performances (SOTAs). Given that not all negatives may be useful (Cai et al., 2020; Jeon et al., 2021), Liu & Chen (2024) makes hard negatives to boost performance, while Lee et al. (2024) utilises soft contrastive learning to weigh sample pairs of varying similarities, both of which reach new SOTAs.

The preceding paragraph outlines a brief summary, and we refer the readers to Section 2 in Lee et al. (2024) and Section 5.3 in Trirat et al. (2024) for a detailed overview of the methods proposed in the past 6 years. These advances have led to increasingly sophisticated methods that mine the contextual information embedded in time series by contrasting similarities. However, the structural details of similarity relations between samples remain to be explored.

### 2.2 STRUCTURE-PRESERVING SSRL

Preserving the original structure of data when mapping into a latent space has been widely and actively researched in manifold learning (for a literature review, Meilă & Zhang, 2024) and graph representation learning (Ju et al., 2024; Khoshraftar & An, 2024). In manifold learning, which is also known as nonlinear dimension reduction, the focus is on revealing the geometric shape of data point clouds for visualisation, denoising, and interpretation. In graph representation learning, the focus is on maintaining the connectivity of nodes in the graph while compressing the data space required for large-scale graphs. Structure-preserving has not yet attracted much dedication to time series data.

Ashraf et al. (2023) provides a literature review on time series data dimensionality reduction, where none of the methods are specifically tailored for time series.

Zooming in within structure-preserving SSRL, there are two major branches respectively focusing on topology and geometry. Topology-preserving SSRL aims to maintain global properties such as clusters, loops, and voids in the latent space; representative models include Moor et al. (2020) and Trofimov et al. (2023) using autoencoders, as well as Madhu & Chepuri (2023) and Chen et al. (2024) with contrastive learning. The other branch is geometry-preserving and focuses more on local shapes such as relative distances, angles, and areas. Geometry-preserving autoencoders include Nazari et al. (2023) and Lim et al. (2024), while Li et al. (2022) and Koishekenov et al. (2023) use contrastive learning. The aforementioned topology and geometry preserving autoencoders are all developed for dimensionality reduction; whereas the combination of contrastive learning and structure-preserving has been explored majorly with graphs.

### 2.3 TRAFFIC INTERACTION SSRL

In line with the conclusions in previous subsections, existing exploration in the context of traffic interaction data and tasks also predominantly relies on autoencoders and graphs. For instance, using a transformer-based multivariate time series autoencoder (Zerveas et al., 2021), Lu et al. (2022) cluster traffic scenarios with trajectories of pairwise vehicles. Then a series of studies investigate masking strategies with autoencoders for individual trajectories and road networks, including Cheng et al. (2023); Chen et al. (2023); Lan et al. (2024).

Leveraging data augmentation, Mao et al. (2022) utilise graphs and contrastive learning to jointly learn representations for vehicle trajectories and road networks. They design road segment positive samples as neighbours in the graph, and trajectory positive samples by replacing a random part with another path having the same origin and destination. Similarly, Zipfl et al. (2023) use a graph-based contrastive learning approach to learn traffic scene similarity. They randomly modify the position and velocity of individual traffic participants in a scene to construct positive samples, with negative samples drawn uniformly from the rest of a training batch. Also using augmentation, Zheng et al. (2024) focuses on capturing seasonal and holiday information for traffic prediction.

## 3 METHODS

This section begins by defining the problem of structure-preserving contrastive learning for spatial time series. Following that, we explain the overall loss function to be optimised, where we propose a dynamic weighing mechanism to balance contrastive learning and structure preservation during training. Then we present the contrastive learning loss for time series, unifying both hard and soft versions in a consistent format. Lastly, we introduce two structure-preserving regularisers, which are respectively adapted to maintain the global and local structure of similarity relations.

### 3.1 PROBLEM DEFINITION

We define the problem for general spatial time series, with traffic interaction as a specific case. Learning the representations of a set of samples $\{\boldsymbol{x}_1, \boldsymbol{x}_2, \cdots, \boldsymbol{x}_N\}$ aims to obtain a nonlinear function $f_{\boldsymbol{\theta}} : \boldsymbol{x} \to \boldsymbol{z}$ that encodes each $\boldsymbol{x}$ into $\boldsymbol{z}$ in a latent space. Let $T$ denote the sequence length of a time series and $D$ the feature dimension at each timestamp $t$. The original space of $\boldsymbol{x}$ can have the form $\mathbb{R}^{T \times D}$, where spatial features are among the $D$ dimensions; or $\mathbb{R}^{T \times S \times D}$, where $S$ represents spatially distributed objects (e.g., sensors or road users). The latent space of $\boldsymbol{z}$ can also be structured in different forms, such as $\mathbb{R}^P$, $\mathbb{R}^{T \times P}$, or $\mathbb{R}^{T \times S \times P}$, where $P$ is the dimension of encoded features.

By contrastive learning, (dis)similar samples in the original space should remain close (far) in the latent space. Meanwhile, by structure preservation, the distance/similarity relations between samples should maintain certain features after mapping into the latent space. We use $d(\boldsymbol{x}_i, \boldsymbol{x}_j)$ to denote the distance between two samples $i$ and $j$, and this also applies to their encoded representations $\boldsymbol{z}_i$ and $\boldsymbol{z}_j$. Various distance measures can be used to define $d$, such as cosine distance (COS), Euclidean distance (EUC), and dynamic time warping (DTW). The smaller the distance between two samples, the more similar they are. Considering the limitation of storage efficiency, similarity comparison is performed in each mini-batch, where $B$ samples are randomly selected.

### 3.2 TRADE-OFF BETWEEN CONTRASTIVE LEARNING AND STRUCTURE PRESERVATION

We define the complete loss function for optimising $f_{\boldsymbol{\theta}}$ as shown in Equation (1). Referring to the simplified loss in Section 1, i.e., $\mathcal{L} = \eta_{\mathrm{CLT}} \cdot \ell_{\mathrm{CLT}} + \eta_{\mathrm{SP}} \cdot \ell_{\mathrm{SP}} + r_{\boldsymbol{\eta}}$, the contrastive learning loss for time series ($\mathcal{L}_{\mathrm{CLT}}$) and structure-preserving loss ($\mathcal{L}_{\mathrm{SP}}$) are modified using the function $x(1 - \exp(-x))$ and correspond to $\ell_{\mathrm{CLT}}$ and $\ell_{\mathrm{SP}}$; $\eta_{\mathrm{CLT}}$, $\eta_{\mathrm{SP}}$, and $r_{\boldsymbol{\eta}}$ depend on two deviation terms $\sigma_{\mathrm{CLT}}$ and $\sigma_{\mathrm{SP}}$, which dynamically change during training.

$$\mathcal{L} = \frac{1}{2\sigma_{\mathrm{CLT}}^2} \mathcal{L}_{\mathrm{CLT}} \left(1 - \exp(-\mathcal{L}_{\mathrm{CLT}})\right) + \frac{1}{2\sigma_{\mathrm{SP}}^2} \mathcal{L}_{\mathrm{SP}} \left(1 - \exp(-\mathcal{L}_{\mathrm{SP}})\right) + \log \sigma_{\mathrm{CLT}} \sigma_{\mathrm{SP}} \tag{1}$$

The modification by $x(1 - \exp(-x))$ serves two purposes: it penalises negative values of $\mathcal{L}_{\mathrm{SP}}$, and stabilises training when either $\mathcal{L}_{\mathrm{CLT}}$ or $\mathcal{L}_{\mathrm{SP}}$ approaches its optimal value. While in computation the value of $\mathcal{L}_{\mathrm{SP}}$ sometimes is below zero, the losses used as $\mathcal{L}_{\mathrm{CLT}}$ and $\mathcal{L}_{\mathrm{SP}}$ in this study all have their theoretical optimal values of zero[1]. As $\mathcal{L}_{\mathrm{CLT}}$ decreases and approaches its optimum zero, the modified term $\ell_{\mathrm{CLT}}$ has a slower decreasing rate when $\mathcal{L}_{\mathrm{CLT}} < 1$. More specifically, the derivative of $x(1 - \exp(-x))$ is $x'(1 - \exp(-x)(1 - x))$, where $x'$ denotes the derivative of $x$ and the multiplier in parentheses decreases from 1 to 0 while $x$ decreases from 1 to 0. This thus stabilises the training when $\mathcal{L}_{\mathrm{CLT}}$ approaches zero, and works the same for $\mathcal{L}_{\mathrm{SP}}$.

Inspired by Kendall et al. (2018), we then weigh the two modified losses by considering their uncertainties. The magnitudes of $\mathcal{L}_{\mathrm{CLT}}$ and $\mathcal{L}_{\mathrm{SP}}$ may vary with different datasets and hyperparameter settings. This variation precludes fixed weights for contrastive learning and structure preservation. We consider the loss values (denoted by $\ell$) as deviations from their optimal values, and learn adaptive weights according to the deviations. Given the optimal value of 0, we assume a Gaussian distribution of $\ell$ with standard deviation $\sigma$, i.e., $p(\ell) = \mathcal{N}(0, \sigma^2)$. Then we can maximise the log likelihood $\sum \log p(\ell) = \frac{1}{2} \sum \left(-\log 2\pi - \log \sigma^2 - \frac{1}{\sigma^2}\ell^2\right)$ to learn $\sigma$. This is equivalent to minimising $\sum \left(\frac{1}{2\sigma^2}\ell^2 + \log \sigma\right)$. When balancing between two losses $\ell_{\mathrm{CLT}}$ and $\ell_{\mathrm{SP}}$ that have deviations $\sigma_{\mathrm{CLT}}$ and $\sigma_{\mathrm{SP}}$, respectively, we need to use Equation (2).

$$\arg\max - \sum \log p(\ell_{\mathrm{CLT}}) p(\ell_{\mathrm{SP}}) \Leftrightarrow \arg\min \sum \left(\frac{1}{2\sigma_{\mathrm{CLT}}^2}\ell_{\mathrm{CLT}} + \frac{1}{2\sigma_{\mathrm{SP}}^2}\ell_{\mathrm{SP}} + \log \sigma_{\mathrm{CLT}} \sigma_{\mathrm{SP}}\right) \tag{2}$$

Replacing $\ell_{\mathrm{CLT}}$ in Equation (2) with $\mathcal{L}_{\mathrm{CLT}} \left(1 - \exp(-\mathcal{L}_{\mathrm{CLT}})\right)$ and $\ell_{\mathrm{SP}}$ with $\mathcal{L}_{\mathrm{SP}} \left(1 - \exp(-\mathcal{L}_{\mathrm{SP}})\right)$, Equation (1) is then derived to be the overall loss. The training process trades-off between $\mathcal{L}_{\mathrm{CLT}}$ and $\mathcal{L}_{\mathrm{SP}}$, as well as between the weight regulariser $r_{\boldsymbol{\eta}} = \log \sigma_{\mathrm{CLT}} \sigma_{\mathrm{SP}}$ and the rest of Equation (1). When $\mathcal{L}_{\mathrm{CLT}}$ is small and $\mathcal{L}_{\mathrm{SP}}$ is large, $\sigma_{\mathrm{CLT}}$ becomes small and $\sigma_{\mathrm{SP}}$ becomes large, which then increases the weight for $\mathcal{L}_{\mathrm{CLT}}$ while reduces the weight for $\mathcal{L}_{\mathrm{SP}}$. The reverse occurs when $\mathcal{L}_{\mathrm{CLT}}$ is large and $\mathcal{L}_{\mathrm{SP}}$ is small. As the weighted sum of $\mathcal{L}_{\mathrm{CLT}}$ and $\mathcal{L}_{\mathrm{SP}}$ increases by larger weights, $\log \sigma_{\mathrm{CLT}} \sigma_{\mathrm{SP}}$ decreases and discourages the increase from being too much. Similarly, if the weighted sum decreases by smaller weights, $\log \sigma_{\mathrm{CLT}} \sigma_{\mathrm{SP}}$ also regularises the decrease.

### 3.3 CONTRASTIVE LEARNING LOSS

In this study, we use the time series contrastive learning loss introduced in TS2Vec (Yue et al., 2022) and its succeeder SoftCLT (Lee et al., 2024) that utilises soft weights for similarity comparison[2]. For each sample $\boldsymbol{x}_i$, two augmentations are created by timestamp masking and random cropping, and then encoded as two representations $\boldsymbol{z}_i'$ and $\boldsymbol{z}_i''$. TS2Vec and SoftCLT losses consider the same sum of similarities for a sample $i$ at a timestamp $t$, as shown in Equations (3) and (4). Equation (3) is used for instance-wise contrasting, which we denote by the subscript $_{\mathrm{inst}}$; Equation (4) is used for time-wise contrasting, denoted by the subscript $_{\mathrm{temp}}$.

$$S_{\mathrm{inst}}(i, t) = \sum_{j=1}^{B} \left(\exp(\boldsymbol{z}_{i,t}' \cdot \boldsymbol{z}_{j,t}'') + \exp(\boldsymbol{z}_{i,t}'' \cdot \boldsymbol{z}_{j,t}')\right) + \sum_{\substack{j=1 \\ j \neq i}}^{B} \left(\exp(\boldsymbol{z}_{i,t}' \cdot \boldsymbol{z}_{j,t}') + \exp(\boldsymbol{z}_{i,t}'' \cdot \boldsymbol{z}_{j,t}'')\right) \tag{3}$$

$$S_{\mathrm{temp}}(i, t) = \sum_{s=1}^{T} \left(\exp(\boldsymbol{z}_{i,t}' \cdot \boldsymbol{z}_{i,s}'') + \exp(\boldsymbol{z}_{i,t}'' \cdot \boldsymbol{z}_{i,s}')\right) + \sum_{\substack{s=1 \\ s \neq t}}^{T} \left(\exp(\boldsymbol{z}_{i,t}' \cdot \boldsymbol{z}_{i,s}') + \exp(\boldsymbol{z}_{i,t}'' \cdot \boldsymbol{z}_{i,s}'')\right) \tag{4}$$

---

[1]We offer a more detailed analysis in Appendix A.1.

[2]The loss function equations in this subsection follow the original papers as closely as possible with minor adjustments based on their open-sourced code.

Equation (5) then shows the TS2Vec loss. We refer the readers to Yue et al. (2022) for more details about the hierarchical contrasting method.

$$\mathcal{L}_{\text{TS2Vec}} = \frac{1}{NT} \sum_i \sum_t \left( \ell^{(i,t)}_{\substack{\text{inst} \\ \text{TS2Vec}}} + \ell^{(i,t)}_{\substack{\text{temp} \\ \text{TS2Vec}}} \right),$$

$$\text{where} \begin{cases} \ell^{(i,t)}_{\substack{\text{inst} \\ \text{TS2Vec}}} = -\log \dfrac{\exp(\boldsymbol{z}'_{i,t} \cdot \boldsymbol{z}''_{i,t}) + \exp(\boldsymbol{z}''_{i,t} \cdot \boldsymbol{z}'_{i,t})}{S_{\text{inst}}(i,t)} \\ \ell^{(i,t)}_{\substack{\text{temp} \\ \text{TS2Vec}}} = -\log \dfrac{\exp(\boldsymbol{z}'_{i,t} \cdot \boldsymbol{z}''_{i,t}) + \exp(\boldsymbol{z}''_{i,t} \cdot \boldsymbol{z}'_{i,t})}{S_{\text{temp}}(i,t)} \end{cases} \tag{5}$$

Similarity comparison in TS2Vec is between two different augmentations for the same sample. This is expanded by SoftCLT to also involve other samples in the same mini-batch. Varying instance-wise and time-wise weights are assigned to different comparison pairs as soft assignments, with Equations (6) and (7). This introduces four hyperparameters, i.e., $\tau_{\text{inst}}$, $\tau_{\text{temp}}$, $\alpha$, and $m$. We use DTW to compute $d(\boldsymbol{x}_i, \boldsymbol{x}_j)$ and set $\alpha = 0.5$, as recommended in the original paper; the other parameters need to be tuned for different datasets. Specifically, $m$ controls the sharpness of time hierarchical contrasting. TS2Vec uses $m = 1$ (constant) and SoftCLT uses $m(k) = 2^k$ (exponential), where $k$ is the depth of pooling layers when computing temporal loss. In this study, we add one more option $m(k) = k + 1$ (linear), and will tune the best way for different datasets.

$$w_{\text{inst}}(i,j) = \frac{2\alpha}{1 + \exp(\tau_{\text{inst}} \cdot d(\boldsymbol{x}_i, \boldsymbol{x}_j)))} + \begin{cases} 1 - \alpha, & \text{if } i = j \\ 0, & \text{if } i \neq j \end{cases} \tag{6}$$

$$w_{\text{temp}}(t,s) = \frac{2}{1 + \exp(\tau_{\text{temp}} \cdot m \cdot |t - s|)} \tag{7}$$

Then Equation (8) shows the SoftCLT loss, where we let $\lambda$ be 0.5 as recommended in the original paper. For a more detailed explanation and analysis, we refer the readers to Lee et al. (2024).

$$\mathcal{L}_{\text{SoftCLT}} = \frac{1}{NT} \sum_i \sum_t \left( \lambda \ell^{(i,t)}_{\substack{\text{inst} \\ \text{SoftCLT}}} + (1 - \lambda) \ell^{(i,t)}_{\substack{\text{temp} \\ \text{SoftCLT}}} \right),$$

$$\text{where} \begin{cases} \ell^{(i,t)}_{\substack{\text{inst} \\ \text{SoftCLT}}} = -\displaystyle\sum_{j=1}^{B} w_{\text{inst}}(i,j) \log \dfrac{\exp(\boldsymbol{z}'_{i,t} \cdot \boldsymbol{z}''_{j,t}) + \exp(\boldsymbol{z}''_{i,t} \cdot \boldsymbol{z}'_{j,t})}{S_{\text{inst}}(i,t)} \\ \qquad - \displaystyle\sum_{\substack{j=1 \\ j \neq i}}^{B} w_{\text{inst}}(i,j) \log \dfrac{\exp(\boldsymbol{z}'_{i,t} \cdot \boldsymbol{z}'_{j,t}) + \exp(\boldsymbol{z}''_{i,t} \cdot \boldsymbol{z}''_{j,t})}{S_{\text{inst}}(i,t)} \\ \ell^{(i,t)}_{\substack{\text{temp} \\ \text{SoftCLT}}} = -\displaystyle\sum_{s=1}^{T} w_{\text{temp}}(t,s) \log \dfrac{\exp(\boldsymbol{z}'_{i,t} \cdot \boldsymbol{z}''_{i,s}) + \exp(\boldsymbol{z}''_{i,t} \cdot \boldsymbol{z}'_{i,s})}{S_{\text{temp}}(i,t)} \\ \qquad - \displaystyle\sum_{\substack{s=1 \\ s \neq t}}^{T} w_{\text{temp}}(t,s) \log \dfrac{\exp(\boldsymbol{z}'_{i,t} \cdot \boldsymbol{z}'_{i,s}) + \exp(\boldsymbol{z}''_{i,t} \cdot \boldsymbol{z}''_{i,s})}{S_{\text{temp}}(i,t)} \end{cases} \tag{8}$$

### 3.4 STRUCTURE-PRESERVING REGULARISERS

We use the topology-preserving loss proposed in (Moor et al., 2020) and the graph-geometry-preserving loss proposed in (Lim et al., 2024) as two structure-preserving regularisers, respectively focusing on the global and local structure of similarity relations. The global structure is preserved for instance-wise comparison, and the local structure is preserved for comparison across temporal or spatial features. In the following, we briefly describe the two losses, and the readers are referred to the original papers for more details.

Equation (9) presents the topology-preserving loss computed in each mini-batch. Here $\boldsymbol{A}$ is a $B \times B$ distance (EUC) matrix between the samples in a batch, and is used to construct the Vietoris-Rips complex; $\boldsymbol{\pi}$ represents the persistence pairing indices of simplices that are considered topologically significant. The superscripts $X$ and $Z$ indicate original data space and latent space, respectively.

$$\mathcal{L}_{\text{topo}} = \frac{1}{2} \left\| \boldsymbol{A}^X \left[ \boldsymbol{\pi}^X \right] - \boldsymbol{A}^Z \left[ \boldsymbol{\pi}^X \right] \right\|^2 + \frac{1}{2} \left\| \boldsymbol{A}^Z \left[ \boldsymbol{\pi}^Z \right] - \boldsymbol{A}^X \left[ \boldsymbol{\pi}^Z \right] \right\|^2 \tag{9}$$

The graph-geometry-preserving loss is also computed per mini-batch, as is shown in Equation (10). This loss measures geometry distortion, i.e., how much $f_{\boldsymbol{\theta}}$ deviates from being an isometry that

preserves distances and angles. The geometry to be preserved of the original data manifold is implied by a similarity graph. To represent temporal and spatial characteristics, instead of using an instance as a node in the graph, we consider the nodes as timestamps or in a spatial dimension such as sensors or road users. Then the edges in the graph are defined by pairwise geodesic distances between nodes.

$$\mathcal{L}_{\text{ggeo}} = \frac{1}{B} \sum_{i=1}^{B} \text{Tr}\left[ \tilde{H}_i \left( L, \tilde{f}_\theta(\boldsymbol{x}_i) \right)^2 - 2\tilde{H}_i \left( L, \tilde{f}_\theta(\boldsymbol{x}_i) \right) \right], \tag{10}$$

where $\tilde{H}_i$ represents an approximation of the Jacobian matrix of $f_{\boldsymbol{\theta}}$. Note that $\tilde{f}_\theta(\boldsymbol{x}_i)$ as the latent representation of $\boldsymbol{x}_i$ needs to maintain the node dimension. For example, if the nodes are considered as timestamps, $\tilde{f}_\theta(\boldsymbol{x}_i) \in \mathbb{R}^{T \times P}$; if the nodes are spatial objects, $\tilde{f}_\theta(\boldsymbol{x}_i) \in \mathbb{R}^{S \times P}$. With a similarity graph defined, then $L$ is the graph Laplacian that is approximated using a kernel matrix with width hyperparameter $h$, which requires tuning for different datasets.

## 4 EXPERIMENTS AND RESULTS

### 4.1 EXPERIMENT SETUP

We compare 6 losses for self-supervised representation learning (SSRL) of time series: TS2Vec, SoftCLT, Topo-TS2Vec, GGeo-TS2Vec, Topo-SoftCLT, and GGeo-SoftCLT. Among the losses, TS2Vec (Yue et al., 2022) and SoftCLT (Lee et al., 2024) are baselines, and the others extend these two with a topology-preserving or a graph-geometry-preserving regulariser. The comparison is then evaluated by downstream task performances using these differently encoded representations. Consequently, the comparison and evaluation serve as an extensive ablation study focusing on the effects of structure-preserving regularisers. Our experiments are conducted with an NVIDIA A100 GPU with 80GB RAM and 5 Intel Xeon CPUs. For fair comparisons, we control the following conditions during experiments: random seed, the space and strategy for hyperparameter search, maximum training epochs, early stopping criteria, and samples used for evaluating local structure preservation.

#### 4.1.1 BASELINES AND DATASETS

The evaluation of performance improvement is on 3 downstream tasks: multivariate time series classification, macroscopic traffic prediction, and microscopic traffic prediction. For every downstream task, we split training/(validation)/test sets following the baseline study and make sure the same data are used across models. Each experiment for a task has two stages, of which the first is SSRL and the second uses the encoded representations to perform classification/prediction. Only the split training set is used in the first stage, with 25% separated as an internal validation set to schedule the learning rate for SSRL.

The classification task is on 28 datasets[3] retrieved from the UEA archive (Bagnall et al., 2018). For each dataset, we set the representation dimension to 320 as used in the TS2Vec and SoftCLT studies, train 6 encoders with the 6 losses, and then classify the encoded representations with an RBF-kernel SVM. For traffic prediction, we use the dataset and model in (Li et al., 2024a) for the macroscopic baseline, and those in (Li et al., 2024b) for the microscopic baseline. The macroscopic traffic prediction uses 40 minutes (2-minute intervals) of historical data in 193 consecutive road segments to predict for all segments in the next 30 minutes. The microscopic traffic prediction forecasts the trajectory of an ego vehicle in 3 seconds, based on the history of up to 26 surrounding road users in the past 1 second (0.1-second intervals). Both traffic prediction baselines use encoder-decoder structures. We first pretrain the encoder with the 6 different losses for SSRL, and then fine-tune the complete model for prediction. The baseline trained from scratch is also compared.

To facilitate clearer analyses when presenting results, we divide the datasets included in the UEA archive into those with spatial features and those without. According to data descriptions in (Bagnall et al., 2018), the UEA datasets are grouped into 6 categories: human activity recognition, motion classification, ECG classification, EEG/MEG classification, audio spectra classification, and other problems. The human activity and motion categories, along with the PEMS-SF and LSST datasets that are categorised as other problems, contain spatial features. We thus consider these as spatial, and the remaining datasets as non-spatial. As a result, each division includes 14 datasets.

---

[3]The UEA archive collects 30 datasets in total. We omitted the two largest, InsectWingbeat and PenDigits, due to limited computation resources.

### 4.1.2 HYPERPARAMETERS

For each dataset, we perform a grid search to find the parameters that minimise $\mathcal{L}_{CLT}$ after a certain number of iterations, where we set a constant learning rate of 0.001. Table 1 shows search spaces of hyperparameters, where bs is abbreviated for batch size and $lr_\eta$ is a separate learning rate for dynamic weights. When searching for best-suited parameters, we first set them as default values, and then follow the search strategy presented in Table 2.

Table 1: Hyperparameter search space.

|  | Default | Search space |
|---|---|---|
| bs | 8 | [8, 16, 32, 64] [a] |
| $lr_\eta$ | 0.05 | [0.01, 0.05] |
| $h$ | 1 | [0.25, 1, 9, 25, 49] |
| $\tau_{temp}$ | 0 | [0.5, 1, 1.5, 2, 2.5] |
| $m$ | constant | [constant, linear, exponential] |
| $\tau_{inst}$ | 0 | [1, 3, 5, 10, 20] |

bs: batch size; $lr_\eta$: learning rate for dynamic weights.

[a]Maximum bs does not exceed train size.

Table 2: Hyperparameter search strategy.

| Stage | bs | $lr_\eta$ | $h$ | $\tau_{temp}$ | $m$ | $\tau_{inst}$ |
|---|---|---|---|---|---|---|
| TS2Vec | △ |  |  |  |  |  |
| Topo-TS2Vec | □ | △ |  |  |  |  |
| GGeo-TS2Vec | □ | △ | △ |  |  |  |
| SoftCLT Phase 1 | ○ |  |  | △ | △ | ○ |
| SofrCLT Phase 2 | △ |  |  | □ | □ | △ |
| Topo-SoftCLT | □ | △ |  | □ | □ | □ |
| GGeo-SoftCLT | □ | △ | △ | □ | □ | □ |

○: default; □: inherited; △: tuned.

The search spaces and strategy can result in up to 63 runs for one dataset. To save searching time, we adjust the number of iterations to be adequate to reflect the progress of loss reduction but limited to prevent overfitting, as our goal is to identify suitable parameters rather than fully train the models. The number of iterations is scaled according to the number of training samples, with larger datasets receiving more iterations.

### 4.1.3 EVALUATION METRICS

Our performance evaluation uses both task-specific metrics and structure-preserving metrics. The former serves to validate performance improvements, while the latter serves to verify the effectiveness of preserving similarity structures. These metrics differ in whether a higher or lower value signifies better performance. To consistently indicate the best method, in the tables presented in the following subsections, the best values are both bold and underlined; the second-best values are bold.

For classification, we use accuracy (Acc.) and the area under the precision-recall curve (AUPRC). To evaluate macroscopic traffic prediction, we use mean absolute error (MAE), root mean squared error (RMSE), the standard deviation of prediction errors (SDEP), and the explained variance by prediction (EVar). Dealing with microscopic traffic, we predict vehicle trajectories and assess the minimum final displacement error (min. FDE) as well as missing rates under radius thresholds of 0.5m, 1m, and 2m ($MR_{0.5}$, $MR_1$, $MR_2$).

As for metrics to evaluate structure preservation, we adopt a combination of those used in (Moor et al., 2020) and (Lim et al., 2024). More specifically, we consider 1) kNN, the proportion of shared k-nearest neighbours according to distance matrices in the latent space and in the original space; 2) continuity (Cont.), one minus the proportion of neighbours in the original space that are no longer neighbours in the latent space; 3) trustworthiness (Trust.), the counterpart of continuity, measuring the proportion of neighbours in the latent space but not in the original space; 4) MRRE, the averaged error in the relative ranks of sample distances between in the latent and original space; and 5) distance matrix RMSE (dRMSE), the root mean squared difference between sample distance matrices in the latent and original space. We calculate these metrics at two scales to evaluate global and local structure preservation. For global evaluation, our calculation is based on EUC distances between samples; for local evaluation, the calculation is based on EUC distances between timestamps for at most 500 samples in a test set.

### 4.2 MULTIVARIATE TIME SERIES CLASSIFICATION

The classification performance on spatial and non-spatial datasets is presented in Table 3. Next to the averaged accuracy, we also include the loss values on test sets to offer more information. More detailed results can be found in Tables A1 and A2 in the Appendix A.2, where we present the classification accuracy with different representation learning losses for each dataset. Then we use Table 4 to more specifically compare the relative improvements induced by adding a topology or graph-geometry preserving regulariser. The relative improvement is the percentage of accuracy difference from the corresponding baseline performance.

Tables 3 and 4 clearly show that structure-preserving improves classification accuracy, not only when time series data involves spatial features, but also when it does not. The relative improvements in Table 4 are higher for non-spatial datasets than for spatial datasets, which is because the datasets without spatial features are more difficult to learn in the UEA archive. As is shown in Table 3, the loss of contrastive learning *decreases* when a structure-preserving regulariser is added for spatial datasets, while *increases* for non-spatial datasets. This implies that preserving similarity structure is well aligned with contrastive learning for spatial datasts, and can even enhance contrastive learning.

Table 3: UEA classification evaluation.

| Datasets | Method | Acc. | AUPRC | $\mathcal{L}_{\text{CLT}}$ | $\mathcal{L}_{\text{SP}}$ |
|---|---|---|---|---|---|
| With spatial features (14) | TS2Vec | 0.848 | 0.872 | 2.943 | |
| | Topo-TS2Vec | 0.851 | 0.876 | 2.264 | 0.085 |
| | GGeo-TS2Vec | 0.856 | 0.881 | 2.200 | 186.9 |
| | SoftCLT | 0.852 | 0.876 | 7.943 | |
| | Topo-SoftCLT | **0.862** | **0.882** | 4.900 | 0.087 |
| | GGeo-SoftCLT | **0.864** | **0.883** | 2.316 | 221.1 |
| Without spatial features (14) | TS2Vec | 0.523 | 0.555 | 8.417 | |
| | Topo-TS2Vec | **0.553** | **0.561** | 11.12 | 0.122 |
| | GGeo-TS2Vec | 0.536 | **0.564** | 15.58 | 957.0 |
| | SoftCLT | 0.508 | 0.532 | 4.714 | |
| | Topo-SoftCLT | 0.496 | 0.534 | 7.328 | 0.124 |
| | GGeo-SoftCLT | **0.537** | 0.549 | 10.09 | 144.7 |

Table 4: Classification accuracy improved by Topo/GGeo regulariser. Comparison is made with corresponding baseline performance.

| Datasets | Improvement by method | Persentage in Acc. (%) | | |
|---|---|---|---|---|
| | | min. | mean | max. |
| With spatial features (14) | Topo-TS2Vec | -4.403 | 0.800 | 16.54 |
| | GGeo-TS2Vec | -3.783 | 1.143 | 10.44 |
| | Topo-SoftCLT | -4.375 | 2.121 | 25.94 |
| | GGeo-SoftCLT | -5.674 | 2.959 | 28.55 |
| Without spatial features (14) | Topo-TS2Vec | -5.263 | 8.852 | 50.00 |
| | GGeo-TS2Vec | -33.33 | 2.083 | 44.44 |
| | Topo-SoftCLT | -33.33 | -0.815 | 50.00 |
| | GGeo-SoftCLT | -20.83 | 18.49 | 166.7 |

Table 5: Structure preservation evaluation over datasets with and without spatial features in the UEA archive.

| Datasets | Method | Local mean between timestamps | | | | | Global mean between all samples | | | | |
|---|---|---|---|---|---|---|---|---|---|---|---|
| | | kNN | Trust. | Cont. | MRRE | dRMSE | kNN | Trust. | Cont. | MRRE | dRMSE |
| With spatial features (14) | TS2Vec | 0.563 | 0.868 | 0.875 | 0.117 | 0.346 | 0.419 | 0.784 | 0.765 | 0.189 | **0.150** |
| | Topo-TS2Vec | **0.569** | **0.873** | 0.878 | **0.114** | 0.344 | 0.418 | 0.783 | 0.764 | 0.190 | 0.154 |
| | GGeo-TS2Vec | **0.569** | **0.873** | **0.881** | **0.114** | **0.341** | 0.418 | 0.781 | 0.762 | 0.190 | 0.157 |
| | SoftCLT | 0.562 | 0.866 | 0.875 | 0.117 | 0.348 | 0.420 | **0.788** | 0.765 | **0.187** | 0.171 |
| | Topo-SoftCLT | 0.564 | 0.869 | 0.877 | 0.115 | 0.344 | **0.421** | 0.784 | **0.767** | 0.188 | 0.153 |
| | GGeo-SoftCLT | **0.571** | **0.875** | **0.883** | **0.111** | **0.337** | **0.425** | **0.790** | **0.768** | **0.185** | **0.149** |
| Without spatial features (14) | TS2Vec | 0.423 | **0.820** | **0.835** | 0.150 | **0.304** | 0.362 | 0.767 | 0.767 | **0.252** | 0.197 |
| | Topo-TS2Vec | 0.424 | **0.820** | 0.831 | 0.151 | **0.308** | 0.356 | 0.763 | 0.767 | 0.254 | **0.191** |
| | GGeo-TS2Vec | 0.420 | **0.820** | 0.832 | 0.151 | 0.310 | **0.365** | **0.769** | **0.771** | 0.253 | **0.189** |
| | SoftCLT | **0.432** | **0.820** | **0.835** | 0.148 | 0.312 | 0.354 | 0.763 | 0.764 | **0.252** | 0.197 |
| | Topo-SoftCLT | 0.426 | 0.818 | 0.834 | **0.148** | 0.312 | 0.361 | **0.768** | 0.768 | 0.254 | 0.205 |
| | GGeo-SoftCLT | **0.430** | **0.822** | **0.835** | **0.147** | 0.315 | 0.355 | 0.761 | 0.762 | 0.257 | 0.203 |

Note: the **best** values are both bold and underlined; the **second-best** values are bold.

The assessment of similarity preservation is presented in Table 5 at both local and global scales. Consistent with the task-specific evaluation, Table 5 shows that structure-preserving regularisation preserves more complete information on similarity relations. The improvements are generally more significant on datasets with spatial features, which makes it more evident that our proposed preservation suits spatial time series data better. Although the comparisons in these tables indicate more notable improvements by preserving graph geometry than preserving topology, we have to note that this does not demonstrate any superiority of one over the other. Different datasets have different characteristics that benefit from preserving global or local structure, and domain knowledge is necessary to determine which could be more effective.

## 4.3 MACROSCOPIC AND MICROSCOPIC TRAFFIC PREDICTION

In Table 6, we present the performance evaluation for both macroscopic and microscopic traffic prediction. This table shows consistent improvements by pretraining encoders with our methods. Notably, single contrastive learning (i.e., TS2Vec and SoftCLT) does not necessarily improve downstream prediction, whereas it does when used together with preserving certain similarity structures. Given that our comparisons are conducted through controlling random conditions, this result effectively shows the necessity of preserving structure when learning traffic interaction representations.

Table 7 then displays the corresponding evaluation on similarity structure preservation, which is obtained by assessing the encoders after fine-tuning for traffic prediction. The results show that

the better performing methods in macro-traffic prediction preserve more global similarity relations between samples; those in micro-traffic prediction, in contrast, preserve more local relations.

Table 6: Macroscopic and microscopic traffic prediction performance evaluation.

| Method | Macroscopic Traffic | | | | Microscopic Traffic | | | |
|---|---|---|---|---|---|---|---|---|
| | MAE (km/h) | RMSE (km/h) | SDEP (km/h) | EVar (%) | min. FDE (m) | MR$_0$.5 (%) | MR$_1$ (%) | MR$_2$ (%) |
| No pretraining | **2.851** | **5.912** | **5.911** | **84.786** | 0.640 | 59.253 | 12.161 | 0.744 |
| TS2Vec | 2.888 | 5.950 | 5.949 | 84.589 | **0.623** | **57.048** | **11.175** | **0.696** |
| Topo-TS2Vec | 2.915 | 5.987 | 5.983 | 84.410 | 0.634 | 58.633 | 11.727 | **0.675** |
| GGeo-TS2Vec | 2.898 | 5.955 | 5.953 | 84.566 | **0.630** | 58.254 | **11.685** | **0.696** |
| SoftCLT | 2.901 | 5.921 | 5.918 | 84.748 | 0.634 | **58.109** | 11.899 | 0.710 |
| Topo-SoftCLT | **2.879** | **5.869** | **5.867** | **85.012** | 0.643 | 59.639 | 12.367 | 0.751 |
| GGeo-SoftCLT | 2.882 | 5.926 | 5.924 | 84.716 | 0.635 | 58.702 | 11.747 | 0.744 |
| Best improvement (%) | 0.000 | 0.715 | 0.745 | 0.266 | 2.644 | 3.721 | 8.102 | 9.259 |

Note: the **best** values are both bold and underlined; the **second-best** values are bold.

Table 7: Structure preservation evaluation of encoders after the fine-tuning in traffic prediction tasks.

| Method | Macroscopic Traffic | | | | | Microscopic Traffic | | | | |
|---|---|---|---|---|---|---|---|---|---|---|
| | kNN | Cont. | Trust. | MRRE | dRMSE | kNN | Cont. | Trust. | MRRE | dRMSE |
| Local mean between timestamps for at most 500 samples | | | | | | | | | | |
| No pretraining | 0.125 | 0.524 | 0.526 | 0.496 | **0.224** | 0.373 | 0.742 | 0.552 | 0.426 | **0.478** |
| TS2Vec | **0.130** | **0.534** | **0.533** | **0.491** | 0.249 | **0.398** | 0.755 | **0.590** | 0.401 | 0.503 |
| Topo-TS2Vec | **0.128** | **0.533** | 0.524 | **0.495** | 0.247 | **0.398** | **0.756** | **0.590** | 0.402 | 0.493 |
| GGeo-TS2Vec | 0.123 | 0.522 | 0.525 | 0.503 | 0.241 | 0.394 | 0.755 | 0.587 | **0.397** | 0.516 |
| SoftCLT | 0.127 | 0.526 | 0.525 | 0.497 | 0.243 | **0.398** | 0.753 | 0.589 | 0.406 | **0.480** |
| Topo-SoftCLT | 0.127 | 0.526 | **0.531** | 0.497 | 0.252 | **0.398** | 0.750 | 0.589 | 0.408 | 0.490 |
| GGeo-SoftCLT | 0.126 | 0.529 | 0.525 | 0.499 | **0.238** | 0.396 | **0.757** | 0.588 | **0.399** | 0.496 |
| Global mean between all samples | | | | | | | | | | |
| No pretraining | **0.316** | **0.949** | **0.969** | **0.031** | **0.364** | 0.218 | 0.937 | 0.920 | 0.049 | 0.141 |
| TS2Vec | 0.268 | 0.936 | 0.959 | 0.040 | 0.406 | 0.234 | **0.954** | 0.927 | **0.042** | 0.142 |
| Topo-TS2Vec | 0.265 | 0.941 | 0.959 | 0.039 | 0.430 | 0.238 | 0.948 | **0.929** | **0.042** | 0.141 |
| GGeo-TS2Vec | 0.268 | 0.940 | 0.958 | 0.040 | 0.405 | **0.239** | **0.963** | 0.922 | **0.040** | **0.140** |
| SoftCLT | 0.293 | **0.943** | 0.964 | **0.035** | 0.386 | 0.215 | 0.907 | 0.901 | 0.066 | 0.147 |
| Topo-SoftCLT | **0.296** | 0.941 | **0.966** | **0.035** | 0.367 | 0.228 | 0.931 | 0.921 | 0.050 | 0.145 |
| GGeo-SoftCLT | 0.286 | 0.941 | 0.963 | 0.037 | **0.365** | **0.243** | 0.945 | **0.927** | 0.044 | **0.140** |

Note: the **best** values are both bold and underlined; the **second-best** values are bold.

Notably, in macroscopic traffic prediction, fine-tuning from scratch maintains the best global similarities. This implies that the specific model architecture might allow for learning similarity structure without pretraining. However, this is not crystal clear with the final evaluation only. In the second part of the next section, we will add two other model architectures for macro-traffic prediction tasks, and visualise the fine-tuning progress to further understand the contribution of structure preservation to downstream task performance.

## 4.4 TRAINING EFFICIENCY

Incorporating structure-preserving regularisation increases computational complexity, and consequently, training time. The magnitude of this increase depends on the data and model that are applied on. With Table 8, we quantify the additional time required for structure preservation and evaluate its impact across diverse model architectures. In prior experiments, we used Convolutional Neural Network (CNN) encoders for the classification on UEA datasets, Dynamic Graph Convolution Network (DGCN, Li et al., 2021) encoder for macroscopic traffic prediction, and VectorNet (Gu et al., 2021) encoder for microscopic traffic prediction. To obtain a more comprehensive evaluation, we include two more Recurrent Neural Network (RNN) models for macroscopic traffic prediction: Long Short-Term Memory (LSTM) and Gated Recurrent Unit (GRU) encoders, paired with simple linear decoders. Results in Table 8 show that preserving structure increases training time by less than 50% in most cases, and suits DGCN particularly well. However, when time sequences are very long (e.g., more than 1,500 steps), the computation of graph-geometry preserving loss becomes intense.

Table 8: Training time per epoch in the stage of self-supervised representation learning.

| Task/data | Encoder | Base (sec/epoch) | TS2Vec | Topo-TS2Vec | GGeo-TS2Vec | SoftCLT | Topo-SoftCLT | GGeo-SoftCLT |
|---|---|---|---|---|---|---|---|---|
| Avg. UEA[a] | CNN | 11.94 | 1.00× | 1.46× | 2.35× | 1.00× | 1.46× | 2.36× |
| MicroTraffic | VectorNet | 128.68 | 1.00× | 1.40× | 1.15× | 1.14× | 1.64× | 1.28× |
| | DGCN | 72.50 | 1.00× | 1.17× | 1.09× | 1.01× | 1.20× | 1.13× |
| MacroTraffic | LSTM | 17.66 | 1.00× | 1.50× | 1.16× | 1.11× | 1.59× | 1.26× |
| | GRU | 15.86 | 1.00× | 1.55× | 1.16× | 1.13× | 1.65× | 1.28× |

[a] Detailed results are referred to Appendix A.2.

Furthermore, we evaluate the fine-tuning efficiency in macroscopic traffic prediction by comparing the convergence speed of methods with and without pretraining. Figure 1 illustrates the influence of structure-preserving pretraining on the fine-tuning progress of different model architectures. For LSTM and GRU, structure preservation consistently enhances prediction performance compared to training from scratch. For DGCN, which is a more sophisticated model tailored for the task, training from scratch is already very effective and only Topo-SoftCLT brings minor improvement.

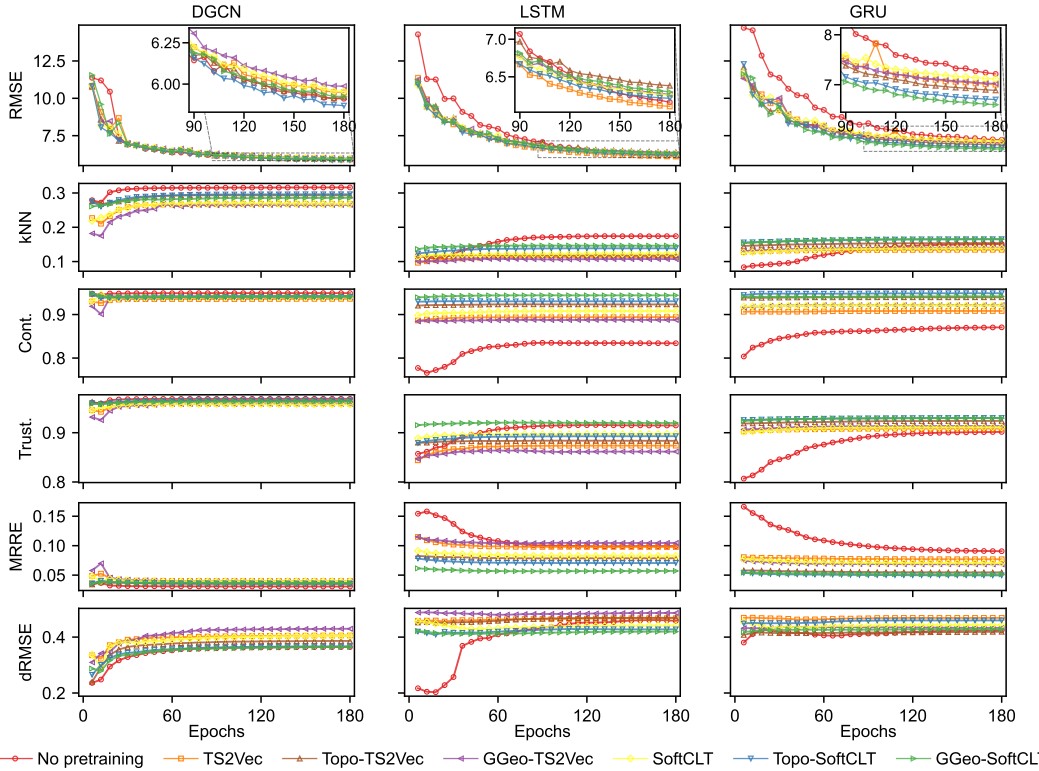

Figure 1: Fine-tuning progress of models pretrained with different losses in macroscopic traffic prediction. Values of the final performance are referred to Table 6 for DGCN, to Tables A5 and A6 for LSTM and GRU.

## 5 CONCLUSION

This paper presents an approach to structure-preserving contrastive learning for spatial time series, where a dynamic mechanism is proposed to adaptively balance contrastive learning and structure preservation. Our methods are experimentally demonstrated to improve the SOTA performance, including for multivariate time series classification in various contexts and for traffic prediction at both macroscopic and microscopic scales. In general, adding structure-preserving regularisation has a limited impact on representation learning efficiency. It can be computationally intensive when the time sequence is long; however, evident performance improvement makes it an acceptable price to pay for utilising the information embedded in time series data. Our experiments (albeit preliminary) also suggest that preserving certain similarity structures may be crucial for downstream task performance, highlighting that the *structural information of similarities in spatio-temporal data remains yet to be exploited*. Given that many real-world practices involve spatial time series, this study and future research based on it can be applied not only to traffic interactions, but also to any that can benefit from preserving specific structures in similarity relations.

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

## A  APPENDIX

### A.1  THEORETICAL OPTIMAL VALUES OF THE LOSSES

For $\mathcal{L}_{\text{TS2Vec}}$, a value of 0 is reached when $z'_{i,t}$ and $z''_{i,t}$ are identical. Similarly, the optimal case of $\mathcal{L}_{\text{SoftCLT}}$ is when the samples with soft assignments close to 1 are identical, while dissimilar samples have soft assignments close to 0. The topology-preserving loss $\mathcal{L}_{\text{topo}}$ is 0 when the topologically relevant distances remain the same in the latent space as in the original space, i.e., $A^X\left[\pi^X\right] = A^Z\left[\pi^X\right]$ and $A^X\left[\pi^Z\right] = A^Z\left[\pi^Z\right]$. Finally, $\mathcal{L}_{\text{ggeo}}$ approximates the distortion measure of isometry and is ideally 0, but can be negative when $\text{Tr}(\tilde{H}_i) < 2$, as the approximation of $\tilde{H}_i$ is kernel-based depending on a hyperparameter $h$.

### A.2  DETAILED RESULTS ON UEA DATASETS

This section provides detailed comparisons of evaluation results for the used 28 datasets in the UEA archive. Tables A1 and A2 present the results of classification accuracy. Tables A3 and A4 present the training time for self-supervised representation learning.

Table A1: Detailed evaluation of classification accuracy on spatial datasets in the UEA archive.

| Dataset | TS2Vec | Topo-TS2Vec | GGeo-TS2Vec | SoftCLT | Topo-SoftCLT | GGeo-SoftCLT |
|---|---|---|---|---|---|---|
| ArticularyWordRecognition | 0.980 | **0.987** | 0.983 | **0.987** | 0.977 | **0.987** |
| BasicMotions | **1.000** | **1.000** | **1.000** | **1.000** | **1.000** | **1.000** |
| CharacterTrajectories | 0.971 | 0.985 | 0.972 | 0.980 | 0.977 | **0.986** |
| Cricket | 0.944 | 0.944 | 0.972 | 0.972 | 0.972 | **0.986** |
| ERing | 0.867 | 0.874 | 0.881 | **0.893** | 0.878 | 0.863 |
| EigenWorms | 0.809 | 0.817 | 0.863 | 0.817 | **0.901** | 0.840 |
| Epilepsy | 0.957 | 0.957 | 0.949 | **0.964** | 0.957 | 0.949 |
| Handwriting | 0.498 | 0.499 | 0.479 | 0.487 | 0.478 | **0.580** |
| LSST | 0.485 | 0.566 | 0.536 | 0.452 | 0.569 | **0.581** |
| Libras | 0.883 | 0.844 | 0.850 | **0.889** | 0.850 | 0.867 |
| NATOPS | 0.917 | 0.917 | 0.933 | 0.922 | 0.917 | **0.944** |
| PEMS-SF | 0.792 | 0.775 | **0.815** | 0.751 | 0.803 | 0.740 |
| RacketSports | 0.908 | 0.914 | 0.914 | **0.928** | 0.908 | 0.875 |
| UWaveGestureLibrary | 0.862 | 0.831 | 0.834 | 0.888 | 0.881 | **0.897** |
| Avg. over spatial datasets | 0.848 | 0.851 | 0.856 | 0.852 | 0.862 | **0.864** |

Table A2: Detailed evaluation of classification accuracy on non-spatial datasets in the UEA archive.

| Dataset | TS2Vec | Topo-TS2Vec | GGeo-TS2Vec | SoftCLT | Topo-SoftCLT | GGeo-SoftCLT |
|---|---|---|---|---|---|---|
| AtrialFibrillation | 0.200 | **0.267** | 0.133 | 0.133 | 0.200 | **0.267** |
| DuckDuckGeese | 0.360 | **0.540** | 0.520 | 0.400 | 0.420 | 0.400 |
| EthanolConcentration | 0.289 | 0.274 | 0.297 | 0.243 | **0.308** | **0.308** |
| FaceDetection | 0.510 | 0.508 | 0.505 | **0.516** | 0.497 | 0.505 |
| FingerMovements | 0.480 | 0.480 | 0.480 | 0.530 | 0.470 | **0.540** |
| HandMovementDirection | 0.324 | **0.405** | 0.257 | 0.324 | 0.230 | 0.257 |
| Heartbeat | 0.751 | **0.761** | 0.717 | 0.756 | 0.737 | 0.732 |
| JapaneseVowels | 0.978 | **0.986** | 0.978 | 0.970 | 0.978 | 0.978 |
| MotorImagery | 0.480 | 0.500 | 0.500 | **0.520** | 0.500 | 0.500 |
| PhonemeSpectra | 0.263 | 0.258 | **0.269** | **0.269** | 0.260 | 0.257 |
| SelfRegulationSCP1 | 0.778 | 0.768 | **0.788** | 0.761 | 0.730 | 0.771 |
| SelfRegulationSCP2 | 0.467 | 0.550 | **0.561** | 0.528 | 0.511 | 0.511 |
| SpokenArabicDigits | 0.973 | **0.976** | 0.966 | 0.964 | 0.968 | 0.957 |
| StandWalkJump | 0.467 | 0.467 | **0.533** | 0.200 | 0.133 | **0.533** |
| Avg. over non-spatial datasets | 0.523 | **0.553** | 0.536 | 0.508 | 0.496 | 0.537 |

Table A3: Detailed representation training time per epoch (unit: s) on spatial datasets in the UEA archive.

| Dataset | TS2Vec | Topo-TS2Vec | GGeo-TS2Vec | SoftCLT | Topo-SoftCLT | GGeo-SoftCLT |
|---|---|---|---|---|---|---|
| ArticularyWordRecognition | 3.799 (1.00×) | 5.61 (1.48×) | 5.863 (1.54×) | 3.772 (0.99×) | 5.77 (1.52×) | 5.983 (1.57×) |
| BasicMotions | 0.475 (1.00×) | 0.685 (1.44×) | 0.709 (1.49×) | 0.457 (0.96×) | 0.687 (1.45×) | 0.711 (1.50×) |
| CharacterTrajectories | 20.640 (1.00×) | 30.863 (1.50×) | 33.32 (1.61×) | 20.652 (1.00×) | 30.948 (1.50×) | 33.18 (1.61×) |
| Cricket | 1.903 (1.00×) | 2.653 (1.39×) | 5.437 (2.86×) | 1.904 (1.00×) | 2.655 (1.40×) | 5.436 (2.86×) |
| ERing | 0.319 (1.00×) | 0.482 (1.51×) | 0.487 (1.53×) | 0.316 (0.99×) | 0.483 (1.51×) | 0.49 (1.54×) |
| EigenWorms | 19.862 (1.00×) | 23.823 (1.20×) | 149.05 (7.50×) | 20.224 (1.02×) | 24.856 (1.25×) | 150.7 (7.59×) |
| Epilepsy | 1.737 (1.00×) | 2.49 (1.43×) | 2.753 (1.58×) | 1.686 (0.97×) | 2.506 (1.44×) | 2.755 (1.59×) |
| Handwriting | 1.875 (1.00×) | 2.771 (1.48×) | 2.959 (1.58×) | 1.88 (1.00×) | 2.775 (1.48×) | 2.987 (1.59×) |
| LSST | 29.786 (1.00×) | 45.273 (1.52×) | 45.162 (1.52×) | 29.859 (1.00×) | 45.216 (1.52×) | 45.154 (1.52×) |
| Libras | 2.081 (1.00×) | 3.142 (1.51×) | 3.142 (1.51×) | 2.085 (1.00×) | 3.135 (1.51×) | 3.141 (1.51×) |
| NATOPS | 1.953 (1.00×) | 2.989 (1.53×) | 2.949 (1.51×) | 2.085 (1.07×) | 3.147 (1.61×) | 3.159 (1.62×) |
| PEMS-SF | 3.413 (1.00×) | 5.069 (1.49×) | 5.38 (1.58×) | 3.415 (1.00×) | 5.064 (1.48×) | 5.399 (1.58×) |
| RacketSports | 1.781 (1.00×) | 2.685 (1.51×) | 2.664 (1.50×) | 1.771 (0.99×) | 2.711 (1.52×) | 2.665 (1.50×) |
| UWaveGestureLibrary | 1.699 (1.00×) | 2.395 (1.41×) | 2.788 (1.64×) | 1.776 (1.05×) | 2.595 (1.53×) | 2.99 (1.76×) |
| Avg. over spatial datasets | 6.523 | 1.46× | 2.12× | 1.00× | 1.48× | 2.15× |

Table A4: Detailed representation training time per epoch (unit: s) on non-spatial datasets in the UEA archive.

| Dataset | TS2Vec | Topo-TS2Vec | GGeo-TS2Vec | SoftCLT | Topo-SoftCLT | GGeo-SoftCLT |
|---|---|---|---|---|---|---|
| AtrialFibrillation | 0.182 (1.00×) | 0.258 (1.42×) | 0.369 (2.03×) | 0.177 (0.97×) | 0.259 (1.42×) | 0.366 (2.01×) |
| DuckDuckGeese | 0.617 (1.00×) | 0.973 (1.58×) | 1.059 (1.72×) | 0.621 (1.01×) | 0.968 (1.57×) | 1.104 (1.79×) |
| EthanolConcentration | 4.939 (1.00×) | 6.655 (1.35×) | 20.128 (4.08×) | 4.89 (0.99×) | 6.664 (1.35×) | 20.182 (4.09×) |
| FaceDetection | 70.709 (1.00×) | 109.6 (1.55×) | 108.83 (1.54×) | 71.104 (1.01×) | 107.523 (1.52×) | 107.092 (1.51×) |
| FingerMovements | 3.826 (1.00×) | 5.67 (1.48×) | 5.706 (1.49×) | 3.779 (0.99×) | 5.671 (1.48×) | 5.716 (1.49×) |
| HandMovementDirection | 2.221 (1.00×) | 3.353 (1.51×) | 4.151 (1.87×) | 2.226 (1.00×) | 3.334 (1.50×) | 4.142 (1.86×) |
| Heartbeat | 2.811 (1.00×) | 4.218 (1.50×) | 5.22 (1.86×) | 2.818 (1.00×) | 4.216 (1.50×) | 5.221 (1.86×) |
| JapaneseVowels | 3.211 (1.00×) | 4.871 (1.52×) | 4.821 (1.50×) | 3.199 (1.00×) | 4.846 (1.51×) | 4.83 (1.50×) |
| MotorImagery | 7.450 (1.00×) | 9.637 (1.29×) | 51.0 (6.85×) | 7.475 (1.00×) | 9.659 (1.30×) | 50.881 (6.83×) |
| PhonemeSpectra | 42.956 (1.00×) | 63.801 (1.49×) | 70.578 (1.64×) | 43.015 (1.00×) | 63.807 (1.49×) | 70.806 (1.65×) |
| SelfRegulationSCP1 | 4.178 (1.00×) | 6.042 (1.45×) | 10.446 (2.50×) | 4.237 (1.01×) | 6.094 (1.46×) | 10.415 (2.49×) |
| SelfRegulationSCP2 | 3.295 (1.00×) | 4.67 (1.42×) | 9.391 (2.85×) | 3.269 (0.99×) | 4.629 (1.40×) | 9.376 (2.85×) |
| SpokenArabicDigits | 96.299 (1.00×) | 143.411 (1.49×) | 131.577 (1.37×) | 86.495 (0.90×) | 125.916 (1.31×) | 129.073 (1.34×) |
| StandWalkJump | 0.304 (1.00×) | 0.404 (1.33×) | 1.68 (5.53×) | 0.31 (1.02×) | 0.4 (1.32×) | 1.7 (5.59×) |
| Avg. over non-spatial datasets | 17.357 | 1.46× | 2.57× | 0.99× | 1.44× | 2.57× |

In addition, to visually show the effect of differently regularised contrastive learning losses on representation, we apply t-SNE to compress the encoded representations into 3 dimensions, as plotted in Figure A1 for the dataset Epilepsy, and Figure A2 for RacketSports. The classes are indicated by colours. We use these two datasets because they are visualisation-friendly, with 4 classes and around 150 test samples.

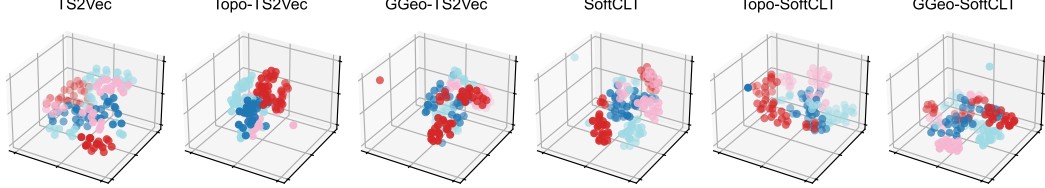

Figure A1: Encoded representations after training with different losses on the test set of Epilepsy.

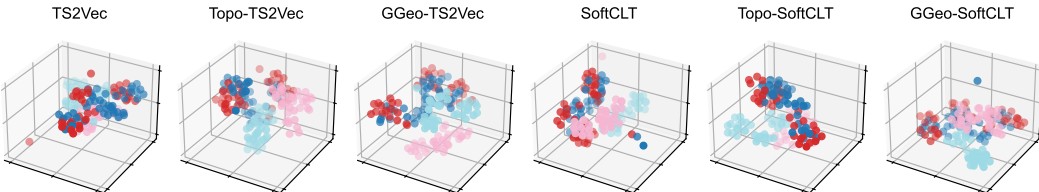

Figure A2: Encoded representations after training with different losses on the test set of RacketSports.

### A.3 DETAILED RESULTS OF MACROSCOPIC PREDICTION WITH LSTM AND GRU

This section provides additional tables presenting the evaluation of the final results using LSTM and GRU in macroscopic traffic prediction. Table A5 shows the task-specific metrics and Table A6 shows the metrics for global structure preservation.

Table A5: Macroscopic traffic prediction evaluation with LSTM and GRU encoders.

| Method | LSTM | | | | GRU | | | |
|---|---|---|---|---|---|---|---|---|
| | MAE (km/h) | RMSE (km/h) | SDEP (km/h) | EVar (%) | MAE (km/h) | RMSE (km/h) | SDEP (km/h) | EVar (%) |
| No pretraining | **3.134** | 6.163 | 6.163 | **83.461** | 3.541 | 7.215 | 7.215 | 77.329 |
| TS2Vec | **3.135** | **6.107** | **6.107** | **83.759** | 3.506 | 7.001 | 7.001 | 78.654 |
| Topo-TS2Vec | 3.172 | 6.266 | 6.266 | 82.901 | 3.495 | 7.022 | 7.022 | 78.528 |
| GGeo-TS2Vec | 3.162 | 6.255 | 6.255 | 82.961 | 3.527 | 7.118 | 7.118 | 77.937 |
| SoftCLT | 3.217 | 6.382 | 6.382 | 82.264 | 3.452 | 6.887 | 6.887 | 79.344 |
| Topo-SoftCLT | 3.176 | 6.214 | 6.214 | 83.183 | **3.410** | **6.703** | **6.702** | **80.438** |
| GGeo-SoftCLT | 3.183 | 6.299 | 6.299 | 82.721 | **3.330** | **6.600** | **6.600** | **81.032** |
| Best improvement (%) | 0.000 | 0.907 | 0.906 | 0.358 | 5.976 | 8.532 | 8.530 | 4.788 |

Note: the **best** values are both bold and underlined; the **second-best** values are bold.

Table A6: Global structure preservation of LSTM and GRU encoders in macroscopic traffic prediction task.

| Method | LSTM | | | | | GRU | | | | |
|---|---|---|---|---|---|---|---|---|---|---|
| | kNN | Cont. | Trust. | MRRE | dRMSE | kNN | Cont. | Trust. | MRRE | dRMSE |
| No pretraining | **0.174** | 0.834 | **0.915** | 0.099 | 0.458 | 0.151 | 0.871 | 0.902 | 0.091 | **0.424** |
| TS2Vec | 0.113 | 0.894 | 0.874 | 0.098 | 0.468 | 0.134 | 0.908 | 0.909 | 0.077 | 0.469 |
| Topo-TS2Vec | 0.107 | 0.888 | 0.862 | 0.105 | 0.486 | 0.141 | 0.921 | 0.914 | 0.068 | 0.433 |
| GGeo-TS2Vec | 0.124 | 0.908 | 0.898 | 0.083 | 0.441 | 0.138 | 0.921 | 0.911 | 0.071 | 0.443 |
| SoftCLT | 0.118 | 0.923 | 0.884 | 0.079 | 0.469 | 0.154 | 0.940 | 0.924 | 0.055 | **0.419** |
| Topo-SoftCLT | 0.139 | **0.931** | 0.893 | **0.071** | 0.428 | **0.165** | **0.949** | **0.930** | **0.050** | 0.460 |
| GGeo-SoftCLT | **0.146** | **0.944** | **0.920** | **0.057** | **0.420** | **0.165** | **0.942** | **0.930** | **0.051** | 0.429 |

Note: the **best** values are both bold and underlined; the **second-best** values are bold.

