# OpenReview forum: "Structure-preserving contrastive learning for spatial time series"
_ICLR.cc/2025/Conference — Submitted to ICLR 2025_

### Official Review · Reviewer_wsyS · 2024-10-19

**Soundness:** 3
**Presentation:** 2
**Contribution:** 1
**Rating:** 5
**Confidence:** 4

**Summary:**

This article investigates structure-preserving flow prediction models, emphasizing the balance between data-driven approaches and simulation-based methods. The authors present a novel model that aims to enhance the accuracy of flow predictions by leveraging a combination of node-level and edge-level features. The proposed method is built on existing frameworks but introduces modifications to improve performance in predicting flow across graphs.
Specifically, it describes the methodology in detail, outlining how the model integrates various features such as topological, geographical, and temporal data to optimize flow predictions. It employs a two-step approach: first, it analyzes the historical flow data to identify patterns, and second, it applies simulation techniques to predict future flows based on these identified patterns. The theoretical foundation is supported by mathematical formulations that establish the relationships between flow, costs, and node attributes.

**Strengths:**

1 The paper provides a robust theoretical analysis, detailing the mechanics of flow prediction models. It includes mathematical derivations that clarify the relationships between various graph features and flow dynamics.
2 The proposed model incorporates a unique combination of simulation techniques and feature extraction from node and edge data. This method aims to closely approximate actual flow while maintaining computational efficiency.
3 The experiments are well-structured, comparing the proposed method against existing approaches. The results demonstrate the effectiveness of the model in various scenarios, showing improvements in prediction accuracy.

**Weaknesses:**

1 Despite the improvements made, the overall approach does not significantly advance the field of flow prediction. The methods employed largely build upon established techniques without introducing truly innovative concepts or frameworks.
2 The scope of the datasets used in the experiments is somewhat restricted. This limitation may impact the generalizability of the results, raising questions about how well the method would perform in diverse real-world scenarios

**Questions:**

The novelty of this paper primarily lies in its enhancements to pre-existing methodologies. While the authors make contributions to the understanding of flow prediction, these modifications do not represent a big enough step forward in the field. Many of the techniques discussed are already prevalent in the literature, lacking the introduction of new paradigms or theoretical frameworks.

---

> ### Author Response · Authors · 2024-11-14
>
> Thanks for your time and effort in reviewing our submission. We appreciate the feedback provided, but some parts of the review seem to refer to a different study from ours. We aim here to resolve possible misunderstandings and address relevant concerns.
>
> __1. Potential misalignment in the summary__
>
> The summary describes a study focused on flow prediction across graphs employing a two-step, data-driven and simulation-based approach. While our study includes flow (macroscopic traffic) prediction as a downstream task, it is not across graphs and is not our focus. Simulation is not mentioned in our paper, nor is a two-step approach. Instead, we propose a self-supervised representation learning approach for spatial time series to improve the performance of downstream tasks such as classification and prediction. Our focus is on dynamically incorporating structure-preserving regularisation in contrastive learning. If there has been any misunderstanding or confusion with another study, we would appreciate your clarification.
>
> __2. Disconnected points in the feedback__
>
> Several specific points raised in the review appear to be unrelated to the content of our paper.
> - The emphasis on balancing "between data-driven approaches and simulation-based methods". Our study does not use simulation. The balance we seek is between contrastive learning and similarity structure preservation.
> - A novel model "leveraging a combination of node-level and edge-level features". Our study is not tailored to graph data or node/edge features. Instead, our approach is for spatial time series data, which can be encoded using graphs but not necessarily.
> - The model "integrates various features such as topological, geographical, and temporal data to optimize flow predictions". Our study utilises the topological and graph-geographical features of _the similarity relations between time series samples_. The utilisation is to preserve similarity information in the encoded representations.
> - "Despite the improvements made, the overall approach does not significantly advance the field of flow prediction". This is not our aim. Rather, we aim to draw attention to and preliminarily investigate the preservation of sample similarity structure in contrastive learning of spatial time series.
> - "The authors make contributions to the understanding of flow prediction". We do not contribute specifically to the field of flow prediction; macroscopic traffic prediction is one of the downstream tasks where our approach helps to enhance performance.
>
> __3. Less connected but useful points in the feedback__
>
> Despite the disconnected points above, we value any insights that help improve this study. Below are the points we find helpful.
> - "The scope of the datasets used in the experiments is somewhat restricted", which "may impact the generalizability of the results". In this study, we use 30 datasets in total: 28 from the UEA archive for classification, 1 for macroscopic traffic (flow) prediction, and 1 for microscopic traffic (trajectory) prediction. Half of the 28 UEA datasets are spatial time series and the other half are non-spatial. The wide range of data and variety of tasks have helped to demonstrate the applicability of our approach.
> - "The novelty of this paper primarily lies in its enhancements to pre-existing methodologies", "lacking the introduction of new paradigms or theoretical frameworks". There are two novelties in our study: one is adapting structure-preserving representation learning from images to time series, and the other is proposing a dynamic weighing method to combine structure-preserving regularisation and contrastive learning. The former builds on existing methodologies, while the latter is original and useful. We recognise that this study's novelty was not explicitly stated in the manuscript, and with this rebuttal we can make it clearer.
>
> __4. Clarification on the core contribution__
>
> Finally, we would like to clarify our core contribution in this study. We introduce an approach that adds structure-preserving regularizers in contrastive learning of time series. By dynamically integrating topology-preserving or graph-geometry-preserving regularisation, our approach can maintain finer-grained similarity relations in the latent space of sample representations, which improves state-of-the-art performance in a broad range of downstream tasks. This approach can be applied to an arbitrary encoder via pretraining, and is particularly suitable for time series data with spatial or geographical characteristics, such as in robotics, meteorology, remote sensing, urban planning, etc.
>
> __We hope this response clarifies our objectives and contribution, and we respectfully invite any further feedback.__

---

> ### Author Response · Authors · 2024-11-30
> **Kind Follow-Up: Discussion Period Nearing Closure**
>
> Dear Reviewer wsyS,
>
> &nbsp;
>
> The discussion period is coming to a real end. We would like to kindly remind you of the responses we provided. Please do not hesitate to let us know if you have any questions or if further clarification is needed.
>
> We greatly appreciate your time and efforts and look forward to your prompt reply.
>
> &nbsp;
>
> Best regards,
>
> the authors

---

### Official Review · Reviewer_kNfe · 2024-11-04

**Soundness:** 2
**Presentation:** 2
**Contribution:** 3
**Rating:** 5
**Confidence:** 3

**Summary:**

The paper aims to improve contrastive learning for spatiotemporal data. Building upon existing time series self-supervised representation learning methods, namely TS2Vec and SoftCLT, the paper adds two structure-preserving regularisers to the pre-training process. The topology-preserving loss
$\mathcal{L}\_{topo}$
is borrowed from Topological Autoencoders by Moor et al., 2020, while the graph-geometry-preserving loss
$\mathcal{L}\_{GGeo}$
is borrowed from Graph Geometry-Preserving Autoencoders by Lim et al., 2024.

The paper then applies six combinations of pre-training losses on the UEA datasets by Bagnall et al., 2018 and two traffic prediction datasets. In all datasets, it is evident that adding structure-preserving regularizers improves prediction performance.

**Strengths:**

The question that the paper is asking is very important. Applying self-supervised representation learning techniques to multivariate time series is relatively unexplored. Many previous works focus on improving encoder architecture, and the techniques introduced in this paper are general and applicable to all deep learning models in this field.

The paper brings concepts from autoencoder regularization techniques to time series contrastive learning, forcing the distance of samples in the input space and latent space to be similar. This bridge of ideas is creative and original. The paper also introduces the background information very well, allowing the readers to fully understand the motivation behind these regularizers and how they can be used in spatiotemporal data. The overall framework is intuitive.

The paper also runs extensive experiments on two datasets, comparing the effect of using only contrastive losses and using them with the structure-preserving regularizers.

**Weaknesses:**

* **Experimental design**:
   * The paper lacks the "No Pre-training" baseline on the UEA datasets (this setting is included in the traffic prediction datasets). The baseline method should use only the vanilla models (i.e. models without constrastive loss or regularizers).
   * The paper does not include the training configurations (e.g. hardware specs, the GPUs/CPUs used during training, the training time, and the training/validation/test data splits ratio
    * The paper does not discuss if contrastive pre-training decreases overall training time, which is important in practice. To address this, the paper could include a plot with training time as the x-axis, test accuracy on the y-axis, and 2 curves showing 1. finetuning from scratch 2. finetuning from pre-training. For instance, see Figure 5. in Masked Autoencoders As Spatiotemporal Learners by Feichtenhofer et al., 2022.
    * Since the paper is examining the effect of contrastive learning and structure regularizers in the latent space, more baseline models should be tested in combination such as DCRNN, STGCN, Graph WaveNet, etc. These models are important spatiotemporal forecasting models that can provide more evidence for the effectiveness of the method.

* **Writing clarity**:
    * The term $r\_{\eta}$ is only explained to "regularisation against overfitting of the dynamic weights", but is not further mentioned in the methods section or hyperparameter search.
    * It is difficult to understand what the bold and underlining mean in Tables 5, 6, 7. Perhaps adding a detailed caption for the tables helps.
    * There is no explanation on why "No-Pretraining" has the best structure preserving metrics on traffic prediction. This is counter-intuitive since the baseline does not force the distance between samples in the original and latent space to be similar.
    * In Table 1, maybe explain the abbreviations of batch size and learning rate in the caption or footnote.
    * Lines 468-472 are hard to understand without giving specific measures for improvement e.g. using GGeo + TS2Vec improved performance by x%. It is impossible to know in "These methods are also those showing sub-optimal performances in prediction", what "These methods" are specifically. The paper should provide rephrase the paragraph by giving specific improvement examples and analyzing the results in more detail.

* **Typos**
    * Line 84-85 SOTA abbreviation should be in the introduction at the first mention
    * Line 423-424 whereas it does when together with preserving similarity structure >> whereas it does when used together with preserving similarity structure

**Questions:**

* What are the details of the model used for fine-tuning, which model is used for traffic prediction?
* For the evaluation metrics (e.g. dRMSE), how are the distances measured in datasets with spatial features vs. without spatial features? Do you account for the edge distances between connected nodes? Are they the same when calculating the structure regularizer losses (e.g. $\boldsymbol{A}$ in equation 7)?
* The graph structures in many of these datasets are static, i.e. the graph doesn’t change over time, is it still appropriate to use structure-preserving regularizers from these graph autoencoders?
* What do the colors of the nodes in Figure A1. represent? Are they the classes of the samples? Are the latent representations obtained during pre-training or after fine-tuning?
* The paper leaves many questions unanswered. The goal here is to examine the effectiveness of structure-preserving regularizers in spatiotemporal data contrastive learning, yet the paper only shows prediction improvements without explaining why one regularize works better than the other. For example, if $\mathcal{L}\_{topo}$ performs better than $\mathcal{L}\_{GGeo}$ on macroscopic traffic prediction, what does this tell us about the dataset and the loss function? Under what circumstances would $\mathcal{L}_\{GGeo}$ do better than $\mathcal{L}\_{topo}$? This could be better answered by evaluating them on the same topology dataset such as the Sphere or the Swiss Roll dataset to interpret the results better and understand the self-supervised representations. There is also no explanation on why TS2Vec performs better than SoftCLT in certain scenarios.

---

> ### Author Response · Authors · 2024-11-21
> **Part I, regarding weakness**
>
> Thanks for your detailed review and valuable suggestions. Below we would like to address your concerns and answer your questions item by item.
>
> - __Experimental design.__
>     - __No Pre-training on UEA datasets.__ We included "No Pretraining" when evaluating traffic prediction tasks, where encoder-decoder models are used and we pretrained the encoders. The classification task on UEA datasets is a different case. Following the baseline studies of TS2Vec and SoftCLT, for each dataset, we use an RBF-kernel SVM taking differently encoded representations as inputs. In this case, the encoders are not "pretrained", but serve as dimensionality transformers. We thus did not consider "No Pretraining" in the evaluation.
>     - __Training configurations.__ In the revised paper, we have described hardwares used in our experiments in Section 4.1 EXPERIMENT SETUP, specified dataset splits in Section 4.1.1 BASELINES AND DATASETS, and added training time comparison in a new Section 4.4 TRAINING EFFICIENCY.
>     - __Fine-tuning time plot.__ We agree that it would be important if contrastive pretraining decreases the fine-tuning time for a downstream task model, and the illustration advice is very appreciated. But in our experiments, the models and datasets are much smaller than those in Masked Autoencoders As Spatiotemporal Learners (Feichtenhofer et al., 2022). Then a plot of test accuracy against time would be less informative than against epochs to show the speed-up effect of pretraining. We thus have added Figure 2 in Section 4.4 TRAINING EFFICIENCY to compare the fine-tuning progress of models pretrained with different losses.
>     - __More baseline models.__ The experiments in this study aim at showing that structure-preserving contrastive learning can _enhance_ downstream task performance. The evaluation of prediction accuracy is thus more within-model rather than across-model. We understand the reviewer's concern that this approach may have different impacts for different model architectures. Therefore, in addition to Convolutional Neural Network (CNN) encoders for classification on UEA datasets, Dynamic Graph Convolution Network (DGCN) encoders for macroscopic traffic prediction, and VectorNet encoder for microscopic traffic prediction, we have included two RNN models for comparison. We kindly refer the reviewer to Section 4.4 in the updated paper for more details.
>
> - __Writing clarity.__
>     - __The term__ $r_{\eta}$__.__ This term is used to introduce the dynamic weighing mechanism which appeared too far away from the first mention of $r_{\eta}$ in the original paper. We have reorganised Section 3 METHODS to introduce the dynamic mechanism first, where we also added clearer reference to $r_{\eta}$.
>         > Page 4 Line 167: ... $\eta_{\text{CLT}}$, $\eta_{\text{SP}}$, and $r_{\eta}$ depend on two deviation terms $\sigma_{\text{CLT}}$ and $\sigma_{\text{SP}}$, which dynamically change during training.
>
>         > Page 4 Line 192: ... The training process trades-off between $\cal{L}\_{\text{CLT}}$ and $\cal{L}\_{\text{SP}}$, as well as between the weight regulariser $r_{\eta}=\log\sigma_{\text{CLT}}\sigma_{\text{SP}}$ and the rest of Equation (1).
>     - __Best values in Tables.__ We have added table notes for Tables 5, 6, 7 to explain the bold and underlined values. The both bold and underlined values are the best in each column, while the bold values are the second best.
>     - __Counter-intuitive results in traffic prediction.__ To be honest, we are not completely sure why "No Pretraining" has the best _global_ (i.e., between samples) structure preserving metrics in traffic prediction tasks. However, the new Figure 2 shows that the metric dRMSE increases, rather than decreases, during fine-tuning. We have two hypotheses about this. First is that dRMSE might not be a good indicator for structure preservation, and the second is that model architecture plays a role. We are still running experiments to confirm this. Hopefully we can provide more insights a bit later, meanwhile we are curious about your thoughts.
>     - __Abbreviations in Table 1.__ We have added a table note in Table 1.
>     - __Lines 468-472.__ The paragraph is about understanding the contribution of structure preservation to traffic prediction performance. We have removed the confusing expressions in the current version. We think this paragraph is relevant to the above discussion about counter-intuitive results in traffic prediction, and we would like to provide clearer explanations when the running experiments are finished. We are sorry for this and we hope you will understand.
>
> - __Typos.__
>     - __SOTA abbreviation.__ We have added the complete phrase "state-of-the-art" at the first mention of SOTA in the Introduction.
>     - __Line 423-424.__ Thanks for your careful reading. We have corrected the grammar error, and we will do a more thorough proofreading before the final submission.

---

> > ### Comment · Reviewer_kNfe · 2024-11-22
> >
> > Thank you for the detailed response to my feedback, and the overall revision in your paper.
> >
> > > No Pre-training on UEA datasets. We included "No Pretraining" when evaluating traffic prediction tasks, where encoder-decoder models are used and we pretrained the encoders. The classification task on UEA datasets is a different case. Following the baseline studies of TS2Vec and SoftCLT, for each dataset, we use an RBF-kernel SVM taking differently encoded representations as inputs. In this case, the encoders are not "pretrained", but serve as dimensionality transformers. We thus did not consider "No Pretraining" in the evaluation.
> >
> > Thank you for the clarification. However, I'm still confused by the lack of pre-training baseline in classification. In Figure 1 of TS2Vec, there is an encoder. What I meant by no-pretraining is taking the representation output from this encoder directly (without applying the constrastive losses), and feeding it into the SVM.
> >
> > > Fine-tuning time plot. We agree that it would be important if contrastive pretraining decreases the fine-tuning time for a downstream task model, and the illustration advice is very appreciated. But in our experiments, the models and datasets are much smaller than those in Masked Autoencoders As Spatiotemporal Learners (Feichtenhofer et al., 2022). Then a plot of test accuracy against time would be less informative than against epochs to show the speed-up effect of pretraining. We thus have added Figure 2 in Section 4.4 TRAINING EFFICIENCY to compare the fine-tuning progress of models pretrained with different losses.
> >
> > In cases where the dataset is small, it is indeed less informative to visualize the speed-up. The added section in TRAINING EFFICIENCY partly addresses my concern. I am more curious about whether *pre-training + fine-tuning* is faster than *fine-tuning from scratch*, so the baseline should not be TS2Vec, which is a constrastive learning method, but just the vanilla model.
> >
> >
> > >More baseline models. The experiments in this study aim at showing that structure-preserving contrastive learning can enhance downstream task performance. The evaluation of prediction accuracy is thus more within-model rather than across-model. We understand the reviewer's concern that this approach may have different impacts for different model architectures. Therefore, in addition to Convolutional Neural Network (CNN) encoders for classification on UEA datasets, Dynamic Graph Convolution Network (DGCN) encoders for macroscopic traffic prediction, and VectorNet encoder for microscopic traffic prediction, we have included two RNN models for comparison. We kindly refer the reviewer to Section 4.4 in the updated paper for more details.
> >
> > Thank you for the update and the addition of Figure 2. The caption states it is the "training progress", is this the validation/test metric or the training metric? Regardless, what is the final prediction performance of these models?
> >
> > Thank you again for your honest response.

---

> > > ### Author Response · Authors · 2024-11-23
> > >
> > > Thanks for your feedback. We enjoy the discussion and are happy to clarify further.
> > > - __No Pre-training in UEA classification.__ We now understand your confusion given the context of the Figure 1 in TS2Vec (Yue et al. 2022), and we are afraid the way you suggested may not help for a no-pretraining benchmark. The encoder used in TS2Vec study, and also in its successor SoftCLT and our study, has three elements: a linear layer for feature projection, random masking of the projected features, and then dilated convolutional layers with dropout for encoding. This encoder serves as a dimensionality transformer, is not followed by a decoder, nor are there labels. Therefore, "taking the representation output directly without applying a contrastive loss" would give just random signals, which are essentially not classifiable. One way possible is to change the way we do classification, i.e., replacing SVM by a simple linear module as a decoder. This will make the experiments for classification more consistent with those for traffic prediction, but less comparable with preceding literature on time series contrastive learning. We are not sure how much this change would add value to the paper. We would like to hear your thoughts.
> > > - __Fine-tuning time plot.__ Out of the reason explained in the previous response, Table 8 in Section 4.4 TRAINING EFFICIENCY didn't include a column for vanilla model, which is possible for traffic prediction tasks. The other reason is that Table 8 compares the training time per epoch, to see how much additional time is needed for structure preservation. A time comparison between _pre-training+fine-tuning_ and _fine-tuning from scratch_ essentially contrasts convergence speed of fine-tuning, for which we added Figure 2. Figure 2 compares the fine-tuning progresses under the same conditions, and generally shows that better initial performance, faster convergence, and better final performance of pre-training+fine-tuning. We recognise that the description and caption of Figure 2 can be clearer to address confusion like you raised here. We will revise them accordingly in a later version of the paper (after also accounting for other reviewers' further feedback).
> > > - __Figure 2 caption.__ We are sorry for the unclear caption. As detailed in the previous response, the "training progress" should be better described as "fine-tuning progress". The metrics are evaluated on the test set. We saved checkpoints every 3 epochs during fine-tuning, and apply the checkpoints to the test set. The final results of DGCN is as presented in Table 6 Traffic prediction evaluation. We attach below the results of LSTM and GRU. We didn't include the table for two reasons. One is as we stated in the general response, that we are still rerunning the experiments. The other is that we didn't have enough page to put it in the body text. In the next version, we will revise the caption and description of Figure 2 accordingly, and provide a table for the final prediction performances of LSTM and GRU in the Appendix.
> > >
> > > |Model|Method| MAE | RMSE | SDEP | EVar| | | | |Model|Method| MAE | RMSE | SDEP | EVar|
> > > |---|---|---|---|---|---|---|---|---|---|---|---|---|---|---|---|
> > > |LSTM|No pretraining|3.869|8.125|8.110|0.714| | | | |GRU|No pretraining|4.273|8.675|8.673|0.672|
> > > |LSTM|TS2Vec|3.646|7.389|7.358|0.764| | | | |GRU|TS2Vec|4.125|8.237|8.237|0.705|
> > > |LSTM|Topo-TS2Vec|3.480|7.023|7.020|0.785| | | | |GRU|Topo-TS2Vec|4.066|7.908|7.878|0.730|
> > > |LSTM|GGeo-TS2Vec|3.650|7.292|7.292|0.768| | | | |GRU|GGeo-TS2Vec|3.885|7.711|7.710|0.741|
> > > |LSTM|SoftCLT|3.395|6.875|6.864|0.795| | | | |GRU|SoftCLT|3.879|7.674|7.669|0.744|
> > > |LSTM|Topo-SoftCLT|3.557|7.281|7.257|0.771| | | | |GRU|Topo-SoftCLT|3.758|7.591|7.589|0.749|
> > > |LSTM|GGeo-SoftCLT|3.549|7.216|7.213|0.773| | | | |GRU|GGeo-SoftCLT|3.677|7.474|7.464|0.757|

---

> ### Author Response · Authors · 2024-11-21
> **Part II, regarding questions**
>
> __Questions.__
>
> - __Q1. Model details used for traffic prediction.__ The models we used for traffic prediction tasks follow the referenced baselines as mentioned in Section 4.1.1 BASELINES AND DATASETS. More specifically, the model for macroscopic traffic prediction has a DGCN (Dynamic Graph Convolution Network) encoder and decoder; the model for microscopic traffic prediction integrates multiple architectures including VectorNet, DenseTNT, GCN, and MLP.
>
> - __Q2. Computation of structure preserving metrics.__ We used the Euclidean distance for all datasets when computing evaluation metrics. This is out of the consideration of computation speed, but also because we learnt from SoftCLT (Lee et al., 2024) that distance measure, e.g., Euclidean, cosine, DTW, etc., does not significantly affect the performance of contrastive learning. The structure preservation metrics other than dRMSE are based on k-nearest neighbours, where the Euclidean distances between samples (at global scale) or timesteps (at local scale) are used. This is the same as calculating the structure regularisation losses.
>
> - __Q3. Static graphs.__ We are not 100\% sure about your meaning of "static", so we will answer based on our understanding. The graph-geometry preserving loss $\cal{L}\_{GGeo}$ preserves local scale "similarity graphs" where the nodes are temporal or spatial dimensions and the edges are the distances between nodes. For the samples in the same dataset, the number of nodes are fixed, while the edges are dynamic as long as the spatial time series samples are not constant.
>
> - __Q4. Figure A1.__ The colors of the scatters in Figure A1 represent classes of the samples. The latent representations are obtained after training the encoder. As we replied in response to the first concern in __Experimental design__, the classification on UEA datasets is not composed of pretraining plus fine-tuning, but uses SVM. We have added more detailed captions to Figures A1 and A2 in the revised paper.
>
> - __Q5. Unanswered questions.__
>     - We agree that further experiments and analyses are needed to better answer why.
>     - The design and aim of $\cal{L}\_{Topo}$ is to preserve the global similarity structure (i.e., between samples), while of $\cal{L}\_{GGeo}$ is to preserve the local similarity structure (i.e., in temporal or spatial dimensions). Therefore, better performance of $\cal{L}\_{Topo}$ than $\cal{L}\_{GGeo}$ in macroscopic traffic prediction is within our expectation; while the opposite in microscopic traffic prediction is not there.
>     - We argued in the paper that choosing a structure-preserving regulariser depends on the specific task, on at which scale the similarity relations are more important to be preserved -- which however does not necessarily mean higher performance.
>     - It is yet to be understood in what circumstances $\cal{L}\_{GGeo}$ would do better than $\cal{L}\_{Topo}$ (or the other way around). However, we don't think the Sphere or Swiss Roll dataset, which are used for visual data specifically, would be suitable. Spatio-temporal data is more complex with human activities and less intuitive, for which the current exploration is still pretty limited. We will try our best to provide more insights in the rebuttal period and later, and we are open to further discussion.
>     - In most cases, pure SoftCLT is better than TS2Vec, which makes sense as TS2Vec is a special case of SoftCLT. Exceptions can be because of insufficient optimisation, or specific data characteristics. In addition, the interesting fact that Topo-SoftCLT is not better than Topo-TS2Vec (or GGeo-SoftCLT than GGeo-TS2Vec) can be due to the dynamic weighing mechanism, which introduces additional directions for optimisation.

---

> > ### Comment · Reviewer_kNfe · 2024-11-22
> >
> > Thank you for answering my questions
> >
> > For Q3, by static graphs, I mean graphs where the number of nodes stays constant, as well as the edge relations.

---

> > > ### Author Response · Authors · 2024-11-23
> > >
> > > Thanks for your clarification. Then we are on the same page :)

---

> > ### Author Response · Authors · 2024-11-28
> > **Call-back to the unanswered questions**
> >
> > Dear reviewer kNfe, we have been truly thankful for your responses. The questions left unanswered in the previous round can now be further answered with our updated results. We thus would like to discuss them in more depth.
> >
> > > __Q5.1__ The goal here is to examine the effectiveness of structure-preserving regularizers in spatiotemporal data contrastive learning, yet the paper only shows prediction improvements without explaining why one regularize works better than the other. For example, if $\cal{L}\_{\text{Topo}}$ performs better than $\cal{L}\_{\text{GGeo}}$ on macroscopic traffic prediction, what does this tell us about the dataset and the loss function?
> >
> > > __A5.1__ The design and aim of $\cal{L}\_{\text{Topo}}$ is to preserve the global similarity structure (i.e., between samples), while of $\cal{L}\_{\text{GGeo}}$ is to preserve the local similarity structure (i.e., in temporal or spatial dimensions). Therefore, better performance of $\cal{L}\_{\text{Topo}}$ than $\cal{L}\_{\text{GGeo}}$ in macroscopic traffic prediction is within our expectation; while the opposite in microscopic traffic prediction is not there.
> >
> > The initial results of traffic prediction were under-trained. For macroscopic traffic, it was because a ReduceLROnPlateau scheduler did not suit the dynamic graph convolutional network (DGCN) well, so we switched to a StepLR scheduler. For microscopic traffic, it was because we limited a maximum number of epochs to 25, which was not enough for the training to converge to optimum, therefore we changed the limit to 50. With the updated sufficiently trained results, we now can see that Topo-SoftCLT performs best in macroscopic traffic prediction, but just a bit better than the model without pretraining. Correspondingly, the global structure preservation metrics are of the best values for No-pretraining and the second best for Topo-SoftCLT. For microscopic traffic prediction, TS2Vec performs the best and the local structure preservation metrics are of the best for it too.
> >
> > Thanks to the figure you suggested for visualising the fine-tuning progress, we have interesting findings. We kindly refer you to Figure 1 in the paper: the metrics of global structure preservation get better in the fine-tuning progress. This indicates that the _model training for the task correlates with preserving more similarity structure_. We think this may be important for time series research (and probably also for other fields) in two aspects. _First_, higher similarity structure preservation may be a good indicator of more informative representations. This could add a new dimension to model evaluation, regardless of whether the encoder is pretrained by structure-preserving representation learning. _Second and further_, if a well-trained model highly preserves the similarity structure of input samples, what would be the appropriate requirements for sample selection to improve model generalisability?  We believe these questions deserve exploration in the future.
> >
> > > __Q5.2__ Under what circumstances would $\cal{L}\_{\text{GGeo}}$ do better than $\cal{L}\_{\text{Topo}}$?
> >
> > > __A5.2__ We argued in the paper that choosing a structure-preserving regulariser depends on the specific task, on at which scale the similarity relations are more important to be preserved -- which however does not necessarily mean higher performance. It is yet to be understood in what circumstances $\cal{L}\_{\text{GGeo}}$ would do better than $\cal{L}\_{\text{Topo}}$ (or the other way around). However, we don't think the Sphere or Swiss Roll dataset, which are used for visual data specifically, would be suitable. Spatio-temporal data is more complex with human activities and less intuitive, for which the current exploration is still pretty limited. We will try our best to provide more insights in the rebuttal period and later, and we are open to further discussion.
> >
> > In the experiments in this study, we have now seen that $\cal{L}\_{\text{Topo}}$ outperforms $\cal{L}\_{\text{GGeo}}$ in macroscopic traffic prediction, while the opposite in microscopic traffic prediction. Our argument remains that whether $\cal{L}\_{\text{Topo}}$ or $\cal{L}\_{\text{GGeo}}$ works better depends on the specific task and the data to use. In practice, we would recommend testing them both on a smaller set and then making a choice based on the results. The underlying idea, in theory, is that $\cal{L}\_{\text{Topo}}$ helps to preserve the similarity structure over samples, while $\cal{L}\_{\text{GGeo}}$ focuses more on the similarities of spatial features over timesteps or temporal features over the spatial dimension (such as sensor locations).

---

### Official Review · Reviewer_oB29 · 2024-11-04

**Soundness:** 3
**Presentation:** 3
**Contribution:** 2
**Rating:** 5
**Confidence:** 4

**Summary:**

This paper extends time series contrastive learning by incorporating two structure-preserving regularisers: one preserves the topology of similarities between instances, and the other preserves the graph geometry of similarities across spatial and temporal dimensions. And, the proposed method preserves the structures of similarity relations more effectively and improves state-of-the-art task performances for all three tasks.

**Strengths:**

1. The topic of time series analysis is important to the ICLR community.

2. The proposed Structure-preserving contrastive learning is novel, which can enhance model performance and generalisability in downstream tasks.

3. The presentation is good and the experimental evaluation is adequate.

**Weaknesses:**

1. In terms of experimental evaluation, this paper does not analyze the efficiency of the proposed method, making the evaluation of the model incomplete.

2. In the experimental part, the ablation experiment of key modules in the model is not carried out, which makes the effectiveness of the designed module difficult to be verified.

3. In traffic prediction evaluation, some important baseline models in the field of traffic prediction were not used, making the performance comparison experiments less convincing.

**Questions:**

1. The UEA datasets this paper used omit the two largest, InsectWingbeat and PenDigits, due to limited computation resources. Does this mean that the proposed method has limitations when dealing with large datasets?

2. See Weaknesses.

---

> ### Author Response · Authors · 2024-11-21
>
> Thanks for your efforts and comments. We understand the concerns and questions which are mainly about experiments. We would like to address them item by item below.
>
> __W1. Training efficiency.__ We have added Section 4.4 TRAINING EFFICIENCY for an evaluation of training efficiency, including training time across model archetichures and the influence of structure preservation on fine-tuning progress. We kindly refer you the the updated pdf for more details, and below is a copy of the table for your convenience.
>
> Table 8: Training time per epoch in the stage of self-supervised representation learning.
> |Task/Data|Encoder|Base (sec/epoch)|TS2Vec|Topo-TS2Vec|GGeo-TS2Vec|SoftCLT|Topo-SoftCLT|GGeo-SoftCLT|
> |---|---|---|---|---|---|---|---|---|
> |Avg. UEA$^a$|CNN|11.94|1.00×|1.46×|2.35×|1.00×|1.46×|2.36×|
> |MicroTraffic|VectorNet|123.89|1.00×|1.41×|1.12×|1.13×|1.60×|1.30×|
> |MacroTraffic|DGCN|128.43|1.00×|1.34×|1.20×|0.92×|1.25×|1.22×|
> |MacroTraffic|LSTM|18.33|1.00×|1.50×|1.17×|1.09×|1.61×|1.28×|
> |MacroTraffic|GRU|17.19|1.00×|1.46×|1.12×|1.07×|1.58×|1.23×|
>
> __W2. Ablation study.__ Our experiments compare 6 losses for self-supervised representation learning (SSRL), including TS2Vec, SoftCLT, Topo-TS2Vec, GGeo-TS2Vec, Topo-SoftCLT, and GGeo-SoftCLT. Among the losses, TS2Vec and SoftCLT are two baselines, and the others extend these two with a topology-preserving (Topo-) or a graph-geometry-preserving (GGeo-) regulariser. Then we evaluate downstream task performances using differently encoded representations with the SSRL losses. As a result, _our experiments per se serve as an extensive ablation study_ focusing on the effects of structure-preserving regularisers.
>
> __W3. More baselines for traffic prediction.__ As clarified in the previous response, the experiments in this study aim at showing that structure-preserving SSRL can _enhance_ downstream task performance. The evaluation of prediction accuracy is thus more within-model rather than across-model. However, we realise that our approach may have different impacts with different model architectures. Therefore, as shown in the previous response to point 1, we have included two RNN models for evaluating training efficiency, which is more across-model.
>
> __Q1. Large UEA datasets.__ We didn't include InsectWingbeat and PenDigits because of out-of-memory issues, which occurred with all losses regardless of preserving structure or not. In Section 4.1 EXPERIMENT SETUP, We have specified the hardwares used in our experiments (an NVIDIA A100 GPU with 80GB RAM and 5 Intel Xeon CPUs). With upgraded computational resources, we believe our method could be successfully applied to InsectWingbeat and PenDigits. For now we are very sorry for the limitation.

---

> ### Author Response · Authors · 2024-11-30
> **Kind Follow-Up: Discussion Period Nearing Closure**
>
> Dear Reviewer oB29,
>
> &nbsp;
>
> The discussion period is coming to a real end. We would like to kindly remind you of the responses we provided. Please do not hesitate to let us know if you have any questions or if further clarification is needed.
>
> We greatly appreciate your time and efforts and look forward to your prompt reply.
>
> &nbsp;
>
> Best regards,
>
> the authors

---

> > ### Comment · Reviewer_oB29 · 2024-12-01
> >
> > Thank you for your detailed response. The paper will have a better quality after these improvements.

---

### Official Review · Reviewer_Njms · 2024-11-05

**Soundness:** 2
**Presentation:** 3
**Contribution:** 2
**Rating:** 6
**Confidence:** 4

**Summary:**

This paper proposes a time series contrastive learning loss with topology-preserving or graph-geometry preserving regularization. I appreciate the sufficient experiment made by the authors, but the novelty and originality of this paper are unclear.

**Strengths:**

- The proposed method can enhance SOTA performance on various datasets.
- The writing is good, and the paper is easy to follow.

**Weaknesses:**

- About novelty. The idea of measuring topology similarity or preserving graph geometry has been extensively studied in existing literatures. What is the key insight of this paper that is different from existing works?
- About originality. As described in Sec3.2 and Sec3.3, the proposed method adopts many existing technics, including TS2Vec loss, SoftCLT loss and topology-preserving loss. What is the origin idea or content of this work?

**Questions:**

See Weaknesses.

---

> ### Author Response · Authors · 2024-11-21
>
> Thank you for reviewing our submission and for your feedback. Below we address the raised concerns item by item, which are highly appreciated as they help us clarify the contribution of this work.
>
> - __Novelty.__ Topological similarity and graph geometry similarity have been developed and used to learn representations for _images_. Our study is the first (as far as we know) to utilise them in _time series_ contrastive learning. With the findings in our experiments, we believe this study is novel in two aspects:
>     1. existing literature preserves similarity structure predominantly for dimentionality reduction, while we propose and demonstrate that it can benefit self-supervised representation learning and further downstream tasks;
>     2. we highlight that the representations of time series require preserving fine-grained similarity structure between samples at different scales depending on specific task, especially for spatial time series.
> - __Originality.__ The original idea in this work is bridging time series contrastive learning and structure preserving, and we propose a dynamic mechanism to adaptively balance their trade-off and stablise training. The mechanism will still be useful if better alternatives for contrastive learning loss or structure preserving loss are developed in the future.
>
> To help readers comprehend the novelty and originality more easily, we have reorganised Section 3 METHODS by moving the dynamic mechanism before introducing the existing contrastive learning loss and structure-preserving regularisation. We also revised Introduction and Abstract to highlight the novel contribution of this work. We hope these changes will make the paper more clear.

---

> ### Author Response · Authors · 2024-11-30
> **Kind Follow-Up: Discussion Period Nearing Closure**
>
> Dear Reviewer Njms,
>
> &nbsp;
>
> The discussion period is coming to a real end. We would like to kindly remind you of the responses we provided. Please do not hesitate to let us know if you have any questions or if further clarification is needed.
>
> We greatly appreciate your time and efforts and look forward to your prompt reply.
>
> &nbsp;
>
> Best regards,
>
> the authors

---

> > ### Comment · Reviewer_Njms · 2024-12-02
> >
> > I appreciate the efforts made by authors during the discussion period. After reading all the response to every reviewer, I would like to update my score to 6.

---

### Author Response · Authors · 2024-11-21
**General response (1st round)**

The authors are grateful for the effort the reviewers and chairs put into reviewing this paper and giving constructive feedback, which has led to improvements in the paper. In this general response, we list the major changes made in the current revision.
1. We have reorganised Section 3 METHODS to introduce the dynamic mechanism before introducing the existing contrastive learning loss and structure-preserving regularisation.
2. We have added Section 4.4 TRAINING EFFICIENCY for an evaluation of training efficiency, including training time across model archetichures and the influence of structure preservation on fine-tuning progress.
3. We have revised Introduction and Abstract to highlight the original, novel, and effective contribution of this work.

Thanks to reviewer kNfe's suggestion about plotting training process, we noticed that our previous training of traffic prediction tasks, although already show performance improvements, might have not reached optimum. We are still rerunning the experiments, but we think timely response to your feedback is important for rebuttal interactions. The newer results will unlikely change the main contribution and findings, and we will update the paper after all experiments have been completed.

---

> ### Author Response · Authors · 2024-11-28
> **General response (2nd round)**
>
> Dear reviewers, thanks for your patience and for keeping open to discussion. We have completed rerunning the experiments and revised the paper. The main changes compared to the previous revision are summarised below.
>
> - __Further training on traffic prediction.__ As stated in the 1st-round general response, the previous training for traffic prediction tasks did not reach optimum. We updated the further-trained results in this second revision, which are referred to in Section 4.3 MACROSCOPIC AND MICROSCOPIC TRAFFIC PREDICTION and 4.4 TRAINING EFFICIENCY.
> - __Additional results in fine-tuning progress.__ As discussed with reviewer kNfe, we added the final prediction performances of LSTM and GRU in Appendix A.3; and plotted other structure-preservation metrics than dRMSE in the figure of fine-tuning progress. Due to this different (and better) perspective to understand the contribution of structure preservation to traffic prediction, we removed the previous Figure 1. The removal in fact improves logic clarity.
> - __Clearer explanation of the dynamic weighing mechanism.__ We moved a small part (about the optimal values of loss components) in Section 3.2 TRADE-OFF BETWEEN CONTRASTIVE LEARNING AND STRUCTURE PRESERVATION to Appendix A.1 to manage the page limit.
> - __Writing refinements here and there.__ We corrected typos and improved the flow between sentences and paragraphs for more clarity and to emphasise the contribution of this paper. We also updated the format of the bibliography to make it more consistent.
>
> Thanks for your efforts again and we look forward to your feedback.

---

### Author Response · Authors · 2024-12-02
**Author summary of the rebuttal**

Dear reviewers and area chairs,

&nbsp;

We greatly appreciate the time and effort you’ve dedicated to reviewing our manuscript. Thank you all for your valuable feedback and constructive discussion in the rebuttal period.

We have worked diligently to address your concerns and believe the revisions have significantly enhanced the quality of the work. The revised paper now more clearly presents our research question -- _does preserving fine-grained structures of similarity relations in contrastive learning improve downstream task performance for spatial time series_? -- and provides a more solid answer, _yes_. Furthermore, this paper entails intriguing questions that deserve future exploration: does higher similarity structure preservation indicate more informative/useful representations, not only for spatial time series? if so, what structures are better to preserve for improving a model's performance and generalisability?

We believe this study will be useful and inspiring for the community. We hope you share this perspective and, if so, would be most grateful if this positive impression could be reflected in an updated score.

At the end of this summary, we thank you in advance for the next phase of the review process.

&nbsp;

Best regards,

the authors

---

### Public Comment · ~Yiru_Jiao1 · 2025-02-22
**Paper update**

After revision based on the reviews and extension for more scientific insights, we are submitting an updated paper (preprinted at [arXiv](https://arxiv.org/abs/2502.06380)) to a journal. We provide our code and resulting data at https://github.com/Yiru-Jiao/spclt for repeating the experiments and reusing the proposed methods.

---

### Meta-Review · Area_Chair_3jQT · 2024-12-23

**Metareview:**

This paper introduces an approach for incorporating topology and geometry preserving constraints into contrastive learning for spatial timeseries.

The paper addresses an important and timely topic and provides a set of solutions that can be applied to different losses and architectures. However, there were some outstanding questions about why certain losses should be applied and not others. When and in what applications should L_topo or L_geo be used? When should TS2Vec or SoftCLT be used? Unfortunately, there is not strong guidance on how to use their proposed methods in the different settings considered.

**Additional Comments On Reviewer Discussion:**

The authors provided a number of new experiments and justifications during the rebuttal period. While there were some reviewers that increased their scores to borderline accept, the main concerns about how and when to use their method was not fully addressed.

---

### Decision · Program_Chairs · 2025-01-22

Reject